# OASIS: Conditional Distribution Shaping for Offline Safe Reinforcement Learning

**Yihang Yao**[*1], **Zhepeng Cen**[*1], **Wenhao Ding**[1], **Haohong Lin**[1], **Shiqi Liu**[1],
**Tingnan Zhang**[2], **Wenhao Yu**[2], **Ding Zhao**[1]
[1] Carnegie Mellon University, [2] Google DeepMind
[*] Equal contribution, {yihangya, zcen}@andrew.cmu.edu

## Abstract

Offline safe reinforcement learning (RL) aims to train a policy that satisfies constraints using a pre-collected dataset. Most current methods struggle with the mismatch between imperfect demonstrations and the desired safe and rewarding performance. In this paper, we mitigate this issue from a *data-centric* perspective and introduce OASIS (cOnditionAl diStributIon Shaping), a new paradigm in offline safe RL designed to overcome these critical limitations. OASIS utilizes a conditional diffusion model to synthesize offline datasets, thus shaping the data distribution toward a beneficial target domain. Our approach makes compliance with safety constraints through effective data utilization and regularization techniques to benefit offline safe RL training. Comprehensive evaluations on public benchmarks and varying datasets showcase OASIS's superiority in benefiting offline safe RL agents to achieve high-reward behavior while satisfying the safety constraints, outperforming established baselines. Furthermore, OASIS exhibits high data efficiency and robustness, making it suitable for real-world applications, particularly in tasks where safety is imperative and high-quality demonstrations are scarce. More details are available at the website https://sites.google.com/view/saferl-oasis/home.

## 1 Introduction

Offline Reinforcement Learning (RL), which aims to learn high-reward behaviors from a pre-collected dataset [1, 2], has emerged as a powerful paradigm for handling sequential decision-making tasks such as autonomous driving [3, 4, 5, 6], and robotics [7, 8, 9, 10, 11]. Although standard offline RL has achieved remarkable success in some environments, many real-world tasks cannot be adequately addressed by simply maximizing a scalar reward function due to the existence of various safety constraints that limit feasible solutions. The requirement for *safety*, or constraint satisfaction, is particularly crucial in RL algorithms when deployed in real-world tasks [12, 13, 14, 15, 16, 17, 18, 19].

To develop an optimal policy within a constrained manifold [20, 21], *offline safe RL* has been actively studied in recent years, offering novel ways to integrate safety requirements into offline RL [22]. Existing research in this area incorporates techniques from both offline RL and safe RL, including the use of stationary distribution correction [23, 24], regularization [25, 26], and constrained optimization formulations [27]. Researchers have also proposed the use of sequential modeling methods, such as the decision transformer [28, 29] and the decision diffuser [30, 31] to achieve advantageous policies and meet safety requirements.

Although these methods show promise, it is difficult to handle state-action pairs that are absent from the dataset, which is known notably as out-of-distribution (OOD) extrapolation issues [31, 32, 33, 34, 35, 36, 37]. To solve this, many works utilize regularization methods to push the policy toward behavior policy to achieve pessimism [35, 36]. However, this approach worsens the situation when the dataset is imbalanced and biased: regularization by imperfect demonstrations such as datasets

38th Conference on Neural Information Processing Systems (NeurIPS 2024).

composed predominantly of low-reward data or containing few safe trajectories collected using unsafe behavior policies. This regularization also leads to another challenge: striking the optimal balance between learning objectives such as task utility efficiency and safety requirements, leading to reward degradation or aggressive behavior [28, 38, 39].

To address these challenges, we introduce a *data-centric* learning paradigm in offline safe RL, OASIS (c**O**ndition**A**l di**S**tribut**I**on **S**haping), which focuses on improving the training dataset quality by steering the offline data distribution to a beneficial target domain as shown in Fig. 1. OASIS distills knowledge from the imperfect dataset, and generates rewarding and safe data using a conditional diffusion model according to the safety preference to benefit offline safe RL training. This *data-centric* approach is parallel to and compatible with general *model-centric* offline safe RL

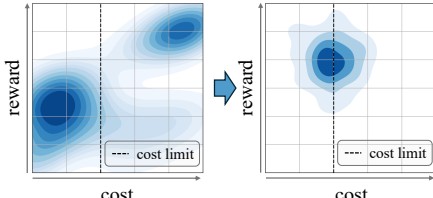

Figure 1: An example of distribution shaping in offline safe RL. We generate a low-cost and high-reward dataset from the original dataset for subsequent RL training.

algorithms which emphasize improvements to the learning algorithm and model architecture. The key contributions are summarized as follows.

**1. Identification of the safe dataset mismatch (SDM) problem in offline safe RL.** We identify the mismatch between the behavior policy and the target policy and investigate the underlying reasons for performance degradation with this condition.

**2. Introduction of the OASIS method to address the SDM problem.** To the best of our knowledge, this is the first successful application of a distribution shaping paradigm within offline safe RL. Our theoretical analysis further provides insights on performance improvement and safety guarantees.

**3. A comprehensive evaluation of our method across various offline safe RL tasks.** The experiment results demonstrate that OASIS outperforms baseline methods in terms of both safety and task efficiency for varying tasks and datasets.

## 2 Related Work

**Offline RL.** Offline RL addresses the limitations of traditional RL, which requires interaction with the environment. The key literature on offline RL includes BCQ [35], which mitigates the extrapolation error using a generative model, and CQL [40], which penalizes the overestimation of Q-values for unseen actions. BEAR [41] further addresses the extrapolation error by constraining the learned policy to stay close to the behavior policy. OptiDICE [24] directly estimates the stationary distribution corrections of the optimal policy, and COptiDICE [23] extends the method to the constrained RL setting. Recent advances have increasingly focused on the use of data generation to improve policy learning. S4RL [42] shows that surprisingly simple augmentations can dramatically improve policy performance. [43] explores leveraging unlabeled data to improve policy robustness, while [44] proposes survival instincts to enhance agent performance in challenging environments. Counterfactual data augmentation is another promising direction in offline RL [45, 46, 47, 48], highlighting the potential of data generation to significantly improve efficiency and effectiveness.

**Safe RL.** Safe RL is formulated as a constrained optimization to maximize reward performance while satisfying the pre-defined safety constraints [20, 49, 50, 51, 52, 53, 54, 55]. Primal-dual framework is one common approach to solve safe RL problem [56, 57, 58, 59, 60, 61, 62, 63]. Another line of work for safe RL is to extend to offline settings, which learn from a fixed dataset to achieve both high reward and constraint satisfaction [64, 29]. Among them, [26, 65, 66, 67] tailor online prime-dual-style algorithms to reduce the out-of-distribution issue in the offline setting. [28, 68] use decision transformer [69] to avoid value estimation and exhibit consistent performances across various tasks. In addition to these *Model-centric* approaches, *Data-centric* approaches, which emphasize improving or optimizing the quality of the dataset used for model training [70, 71, 72, 73], have gained more attention in recent studies. While some previous work proposed methods for learning from safe demonstration [28, 74] or relabeling data to achieve conservativeness [28], how to systematically curate datasets for offline safe learning remains a largely unexplored area.

**Diffusion Models for RL.** Diffusion models have recently gained attention in RL for their capabilities in planning and data generation [75, 76, 77, 78]. Specifically, Diffuser [79] uses a diffusion process to plan the entire trajectory in complex environments. [80] extends this to the Decision Diffuser,

which conditions the diffusion process on specific goals and rewards to improve decision-making. SafeDiffuser [81] and FISOR [31] enhance safety by ensuring the planned trajectories satisfying constraints. Combined with the data augmentation capability of diffusion models, AdaptDiffuser [82] achieves state-of-the-art results on offline RL benchmarks. [83] proposes Synthetic Experience Replay, leveraging diffusion models to create synthetic experiences for more efficient learning. [84] demonstrates that diffusion models are effective planners and data synthesizers for multi-task RL, showcasing their versatility and efficiency. In this work, we investigate the power of diffusion models for safe RL, where the balance between reward and cost presents further complexities.

## 3 Problem Formulation

### 3.1 Safe RL with Constrained Markov Decision Process

We formulate Safe RL problems under the Constrained Markov Decision Process (CMDP) framework [85]. A CMDP $\mathcal{M}$ is defined by the tuple $(\mathcal{S}, \mathcal{A}, \mathcal{P}, r, c, \gamma, \mu_0)$, where $\mathcal{S} \in \mathbb{R}^m$ is the state space, $\mathcal{A} \in \mathbb{R}^n$ is the action space, $\mathcal{P} : \mathcal{S} \times \mathcal{A} \times \mathcal{S} \to [0, 1]$ is the transition function, $r : \mathcal{S} \times \mathcal{A} \times \mathcal{S} \to \mathbb{R}$ is the reward function, $c : \mathcal{S} \times \mathcal{A} \times \mathcal{S} \to \mathbb{R}_{\geq 0}$ is the cost function, $\gamma$ is the discount factor, and $\mu_0 : \mathcal{S} \to [0, 1]$ is the initial state distribution. Note that this work can also be applied to multiple-constraint tasks, but we use a single-constraint setting for easy demonstration. A safe RL problem is specified by a CMDP and a constraint threshold $\kappa \in [0, +\infty)$. Denote $\pi \in \Pi : \mathcal{S} \times \mathcal{A} \to [0, 1]$ as the policy and $\tau = \{(s_1, a_1, r_1, c_1), (s_2, a_2, r_2, c_2), \dots\}$ as the trajectory. The stationary state-action distribution under the policy $\pi$ is defined as $d^\pi(s, a) = (1 - \gamma) \sum_t \gamma^t \Pr(s_t = s, a_t = a)$. The reward and cost returns are defined as $R(\tau) = \sum_\tau r$, and $C(\tau) = \sum_\tau c$. The value function is $V_{\mathbf{f}}^\pi(\mu_0) = \mathbb{E}_{\tau \sim \pi, s_0 \sim \mu_0}[\sum_{t=0}^\infty \gamma^t \mathbf{f}_t], \mathbf{f} \in \{r, c\}$, which is the expectation of discounted return under the policy $\pi$ and the initial state distribution $\mu_0$. The goal of safe RL is to find the optimal policy $\pi^*$ that maximizes the expectation of reward return while constraining the expectation of cost return to the threshold $\kappa$:

$$\pi^* = \arg\max_\pi \mathbb{E}_{\tau \sim \pi}\big[R(\tau)\big], \quad s.t. \quad \mathbb{E}_{\tau \sim \pi}\big[C(\tau)\big] \leq \kappa. \tag{1}$$

### 3.2 Regularized offline safe RL

For an offline safe RL problem, the agent can only access a pre-collected dataset $\mathcal{D} = \cup_i \mathcal{D}_i$, where $\mathcal{D}_i \sim \pi_i^B$ is collected by the behavior policy $\pi_i^B \in \Pi^B$. To solve the problem in Eq. (1), we convert the constraint optimization problem into an unconstrained form:

$$(\pi^*, \lambda^*) = \arg\max_\lambda \min_\pi \mathcal{J}(\pi, \lambda), \quad \mathcal{J}(\pi, \lambda) = -\mathbb{E}_{\tau \sim \pi} R(\tau) + \lambda(\mathbb{E}_{\tau \sim \pi} R(\tau) - \kappa). \tag{2}$$

The primal-dual-based algorithm solves the optimal policy $\pi^*$ and the dual variable $\lambda^*$ by updating $(\pi, \lambda)$ iteratively [86, 23, 66]. In offline safe RL tasks, a regularization term is usually introduced to prevent the action OOD issue [87], that is, the objective is converted to:

$$(\pi^*, \lambda^*) = \arg\max_\lambda \min_\pi \mathcal{J}_{\text{off}}(\pi, \lambda), \quad \mathcal{J}_{\text{off}}(\pi, \lambda) = \mathcal{J}(\pi, \lambda) + wL(\pi, \pi^B), \tag{3}$$

where $w > 0$ is a constant weight, $L(\pi, \pi^B)$ is a regularization term and $\pi^B$ is the empirical behavior policy and can be viewed as a mixture of $\{\pi_i^B\}$. Practically, regularization is formulated as the MSE regularization [88] or the evidence lower bound regularization [35, 41]. In offline safe RL, there are two main challenges: (1) **Distribution shift** [25]. The agent has poor generalizability when facing OOD state-action pairs during online evaluation; and (2) **Efficiency-safety performance balancing** [28]. The agent tends to be over-conservative or aggressive when overestimating or underestimating the safety requirements.

## 4 Method

In this section, we first identify the *safe dataset mismatch* (SDM) problem, which leads to performance degradation when solving the regularized offline safe RL objective in Eq. (3). Then we present the proposed OASIS (c**O**ndition**A**l di**S**tribut**I**on **S**haping) method to solve this problem. In contrast to the *model-centric* safe RL approaches that focus on optimizing policy update process and network architecture, OASIS is a *data-centric* learning method that aims to improve the quality of the dataset, thus benefiting offline safe RL training. The OASIS method utilizes the diffusion model to realize conditional distribution shaping, solving the challenges mentioned above, thus benefiting offline safe RL training. Following the proposed algorithm, we provide a theoretical guarantee of the safety performance of the policy learned in this paradigm.

## 4.1 Safe Dataset Mismatch Problem

The regularized offline safe RL objective in Eq. (3) pushes policy to behavior policy to prevent action OOD issues [36]. When given an imperfect dataset, the state-action distribution deviates from the optimal distribution, and the SDM problem arises: if the behavior policy is too conservative with low costs and low rewards, it leads to task efficiency degradation; if the behavior policy is too aggressive with high costs and high rewards, it leads to safety violations. To further investigate the SDM problem and the effect of dataset distribution on offline safe RL, we define the **tempting dataset** and **conservative dataset**, which are based on tempting and conservative policies:

**Definition 1** (Tempting policy [51] and conservative policy). The tempting policy class is defined as the set of policies that have a higher reward return expectation than the optimal policy, and the conservative policy class is defined as the set of policies that have lower reward and cost return expectations than the optimal policy:

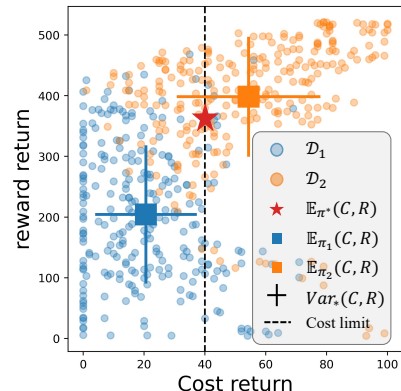

Figure 2: $\mathcal{D}_1$ is a conservative dataset, and $\mathcal{D}_2$ is a tempting dataset. Each point represents $(C(\tau), R(\tau))$ of a trajectory $\tau$ in the dataset.

$$\Pi^T := \{\pi : V_r^\pi(\mu_0) > V_r^{\pi^*}(\mu_0)\}, \quad \Pi^C := \{\pi : V_r^\pi(\mu_0) < V_r^{\pi^*}(\mu_0), V_c^\pi(\mu_0) < V_c^{\pi^*}(\mu_0)\}. \quad (4)$$

Intuitively, a tempting policy is a more rewarding but less safe policy than the optimal one, and a conservative policy is with lower cost but less rewarding. According to these policies, we define two types of datasets:

**Definition 2** (Tempting and conservative dataset). For an offline dataset $D_i \sim \pi_i$, if $\pi_i \in \Pi^B \cap \Pi^T$, then the dataset is tempting; if $\pi_i \in \Pi^B \cap \Pi^C$, then the dataset is conservative.

Staying within the tempting dataset distribution results in tempting (unsafe) behavior, while staying within the conservative dataset distribution causes reward degradation. A theoretical analysis of performance degradation due to the SDM problem is presented in Sec. 4.4. Fig. 2 illustrates examples of both conservative and tempting datasets.

It is important to note that tempting and conservative datasets are prevalent in offline safe RL since optimal policies are rarely available for data collection. The SDM problem is a distinct feature of offline safe RL, indicating that training the policy on either tempting or conservative datasets will violate safety constraints or result in sub-optimality, both of which are undesirable. Therefore, addressing the SDM problem is essential for the development of regularized offline safe RL algorithms.

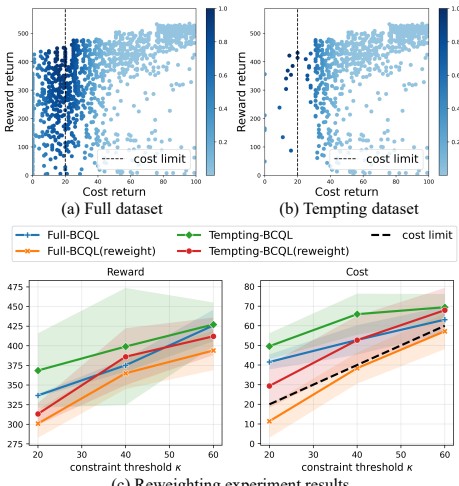

Figure 3: (a) Reweighting in the dataset with comprehensive coverage. (b) Reweighting in the tempting dataset. (c) Performance evaluation with different weights and datasets.

## 4.2 Mitigating the Safe Dataset Mismatch

Inspired by recent research works on *data-centric* learning [70], which emphasizes the quality of data used for training the agent, we focus on enhancing the data pipeline to ensure the agent learns effectively instead of modifying the RL algorithm or model architecture. We propose to use the distribution shaping of the dataset to mitigate the SDM problem, that is, generating a new dataset $\mathcal{D}_g$ by reshaping the original data distribution. As shown in Fig. 1, the key idea is to adjust the dataset distribution towards the optimal distribution under $\pi^*$, reducing the distribution discrepancy and mitigating the SDM problem, thus both mitigating the action OOD issue and balancing efficiency and safety in offline safe RL.

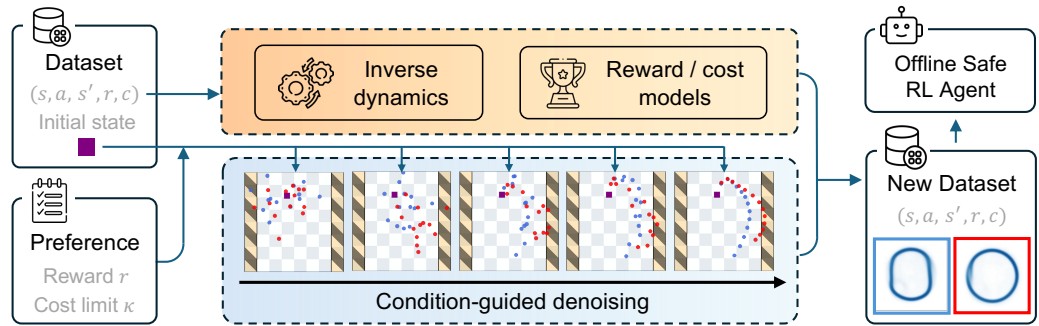

Figure 4: OASIS: a *data-centric* approach for offline safe RL. Conditioned on the human preference, OASIS first curates an offline dataset with a conditioned diffusion data generator and learned labeling models, then trains safe RL agents with this generated dataset.

Among the *data-centric* algorithms, Dataset reweighting (weighted sampling) which assigns different sampling weights to data points, is a straightforward way to shape the data distribution [39, 89]. In the offline RL domain, researchers proposed methods to assign high weights to data that achieve high rewards and superior performance in many RL tasks [39, 89].

To validate this idea, we deploy a Boltzmann energy function considering both the reward and the cost for the reweighing strategy to solve the problem (see Appendix B for details). The experimental results, shown in Fig. 3, validate the effectiveness of this distribution shaping method when the coverage of the dataset is complete.

However, for a more general case where we can only access the low-quality dataset (e.g., tempting datasets in Fig. 3), simply performing data reweighting does not work well due to the absence of necessary data. Thus, we propose to use a conditional generative model for more flexible distribution shaping, which generates new data by stitching sub-optimal trajectories for offline training.

### 4.3 Constraint-Conditioned Diffusion Model as Data Generator

To overcome the limitation of reweighing methods, we propose using diffusion models to generate the dataset that fits the target cost limit to achieve distribution shaping. In the following, we introduce the details of the generator training and dataset generation phases.

**Training.** In previous works [82, 79], the trajectory planning in offline RL can be viewed as the sequential data generation: $\boldsymbol{\tau} = [s_0, s_1, \ldots, s_{L-1}]$, where $\boldsymbol{\tau}$ is a subsequence of trajectory with length $L$. Denote $\boldsymbol{x}_k(\boldsymbol{\tau})$ and $\boldsymbol{y}(\boldsymbol{\tau})$ as the $k$-step denoising output of the diffusion model and the denoising conditions such as reward and cost returns, respectively. Then the forward diffusion process is to add noise to $\boldsymbol{x}_k(\boldsymbol{\tau})$ and gradually convert it into Gaussian noise:

$$q\left(\boldsymbol{x}_k(\tau) \mid \boldsymbol{x}_{k-1}(\tau)\right) := \mathcal{N}\left(\boldsymbol{x}_k(\tau); \sqrt{1 - \beta_k}\boldsymbol{x}_{k-1}(\tau), \beta_k \boldsymbol{I}\right), \ k = 1, ..., K \tag{5}$$

where $\beta_k$ is a pre-defined beta schedule, $K$ is the total denoising timestep. Then the trainable denoising step aims at gradually converting the Gaussian noise back to a valid trajectory:

$$p_\theta\left(\boldsymbol{x}_{k-1}(\tau) \mid \boldsymbol{x}_k(\tau), \boldsymbol{y}(\boldsymbol{\tau})\right) := \mathcal{N}\left(\boldsymbol{x}_{k-1}(\tau) \mid \mu_\theta\left(\boldsymbol{x}_k(\tau), \boldsymbol{y}(\tau), k\right), \Sigma_k\right), \tag{6}$$

where $\theta$ is the trainable parameter. We use a simplified surrogate loss [90] for optimization:

$$\mathcal{L}_{\text{denoise}} := \mathbb{E}_{\boldsymbol{x}_0(\tau) \sim q, \epsilon \sim \mathcal{N}(\boldsymbol{0}, \boldsymbol{I})}\left[\|\epsilon - \epsilon_\theta\left(\boldsymbol{x}_k(\tau), \boldsymbol{y}(\tau), k\right)\|^2\right]. \tag{7}$$

In this work, we use the classifier-free guidance [91] for conditional data generation. The condition $\boldsymbol{y}(\tau)$ in Eq. (6) and Eq. (7) is set to $\boldsymbol{y}(\tau) = [C(\tau), R(\tau)]$. Thus, the denoising process depends on the target reward and cost returns of the planned subtrajectory. During training time, the diffusion model learns both an unconditional denoising core $\epsilon_\theta\left(\boldsymbol{x}_k(\tau), \varnothing, k\right)$ and a conditional denoising core $\epsilon_\theta\left(\boldsymbol{x}_k(\tau), \boldsymbol{y}(\tau), k\right)$. We adopt masking [80] for the training to zero out the condition of one training trajectory and categorize it as the $\varnothing$ class with probability $0 < p < 1$. Within the given raw dataset, we also train an inverse dynamics model $\hat{f} : \mathcal{S} \times \mathcal{S} \to \mathcal{A}$, and reward and cost models $\hat{r}(s, a, s'), \hat{c}(s, a, s') : \mathcal{S} \times \mathcal{A} \times \mathcal{S} \to \mathbb{R}$ for labeling.

**Generation.** After obtaining a trained model, the next step is to generate a new dataset following the conditions. For diffusion model inference, the denoising core $\epsilon_\theta(\boldsymbol{x}_k(\tau), \boldsymbol{y}, k)$ is calculated by:

$$\epsilon_\theta(\boldsymbol{x}_k(\tau), \boldsymbol{y}(\tau), k) = \epsilon_\theta(\boldsymbol{x}_k(\tau), \varnothing, k) + w_\alpha(\epsilon_\theta(\boldsymbol{x}_k(\tau), \boldsymbol{y}_c(\tau), k) - \epsilon_\theta(\boldsymbol{x}_k(\tau), \varnothing, k)), \quad (8)$$

where $w_\alpha > 0$ is a constant guidance scale and $\boldsymbol{y}_c := [\hat{C}, \hat{R}]$ is the generation condition, we set $\hat{C} \leq \kappa$ to align safety preference. For guided generation, we fix the initial state, which means that we replace the initial state of each $k$-step noised trajectory as $\boldsymbol{x}_k[0] = \boldsymbol{x}_0[0]$[1].

After generating one subtrajectory $\tau_g = \boldsymbol{x}_0$, we can get the state and action sequence $s_g = \tau_g[:-1], s_g' = \tau_g[1:], a_g = \hat{f}(s_g, s_g')$, then label the data $r_g = \hat{r}(s_g, a_g, s_g'), c_g = \hat{c}(s_g, a_g, s_g')$. Finally, we get a generated dataset $\mathcal{D}_g = \{s_g, a_g, s_g', r_g, c_g\}$ with $|\tau_g| - 1$ transition pairs. With this new dataset, we can further train offline safe RL agents.

In this work, we consider `BCQ-Lag` [35, 86] as the base offline safe RL algorithm. The process of generating one subtrajectory $\tau_g$ is summarized in Algorithm 1. More details of the implementation are available in Appendix C.

---

**Algorithm 1:** OASIS (dataset generation)

**Input:** Raw dataset $\mathcal{D}$, safety threshold $\kappa$
**Output:** Generated sub-trajectory $\tau_g$
1: Sample a initial state: $s \sim \mathcal{D}$
2: Get initial noisy sub-trajectory:
   $\boldsymbol{x}_k = [s, s_1, ..., s_{L-1}], s_i \sim \mathcal{N}(\boldsymbol{0}, \boldsymbol{I})$
3: Determine the condition $\boldsymbol{y}_c$;
4: **for** $k = K, ..., 1$ **do**
5:    Calculate $\epsilon(\boldsymbol{x}_k, \boldsymbol{y}_c, k)$ (8);
6:    Inverse sampling sequence $\boldsymbol{x}_{k-1}$ (6)
7: **end for**
8: Get actions, rewards, and costs from $\boldsymbol{x}_0$;
9: **Return:** trajectory $\tau_g$

---

### 4.4 Theoretical analysis

We first investigate how the distribution mismatch degrades the policy performance on constraint satisfaction. Suppose that the maximum one-step cost is $C_{\max} = \max_{s,a} c(s, a)$. Based on Lemma 6 in [92] and Lemma 2 in [93], the performance gap between the policy $\pi$ learned with the dataset $\mathcal{D}$ and the optimal policy is bounded by

$$|V_c^\pi(\mu_0) - V_c^*(\mu_0)| \leq \frac{2C_{\max}}{1-\gamma} D_{\mathrm{TV}}(d^\mathcal{D}(s)\|d^*(s)) + \frac{2C_{\max}}{1-\gamma} \mathbb{E}_{d^*(s)}[D_{\mathrm{TV}}(\pi(a|s)\|\pi^*(a|s))], \quad (9)$$

where $d^\mathcal{D}(s), d^*(s)$ denote the stationary state distribution of the dataset and optimal policy. The proof is given in Appendix A.1. Therefore, a significant mismatch between the dataset and the optimal policy results in both a substantial state distribution TV distance and a large policy shift from the optimal one, which can cause notable performance degradation, especially when the offline RL algorithm enforces the learned policy to closely resemble the behavior policy of the offline data.

Then we provide a theoretical analysis of how our method mitigates this mismatch issue by shrinking the distribution gap, which provides a guarantee of the safety performance of the regularized offline safe RL policy. Let $d_g(s|\boldsymbol{y})$ denote the state marginal of the generated data with condition $\boldsymbol{y}$. We first make the following assumptions.

**Assumption 1** (Score estimation error of the state marginal)**.** There exists a condition $\boldsymbol{y}^*$ such that the score function error of the state marginal is bounded by

$$\mathbb{E}_{d^*(s)}\|\nabla_s \log d_g(s|\boldsymbol{y}^*) - \nabla_s \log d^*(s)\| \leq \varepsilon_{\mathrm{score}}^2, \quad (10)$$

where $d^*(s)$ is the stationary state distribution induced by the optimal policy $\pi^*$.

This assumption is also adopted in previous work [94, 95]. For simplicity, we omit the condition $\boldsymbol{y}^*$ in the following analysis and use $d_g(s), d_g(s, a)$ to denote the generated state or the state-action distribution with condition $\boldsymbol{y}^*$. As we use inverse dynamics $f(a|s, s')$ to calculate actions based on the generated state sequence, the quality of the dataset is also determined by the inverse dynamics. Therefore, we further make the following assumption.

**Assumption 2** (Error of inverse policy)**.** The error of action distribution generated by the inverse dynamics is bounded by

$$\mathbb{E}_{d^*(s)}[D_{\mathrm{KL}}(\hat{\pi}_{\mathrm{inv}}(\cdot|s)\|\pi^*(\cdot|s))] \leq \varepsilon_{\mathrm{inv}}, \quad (11)$$

---

[1]This training/generation formula fixing initial state in the sequence is adopted from Decision Diffuser [80]. However, we also found that fixing the initial state is unnecessary for our dataset curation and conditional distribution shaping task. Keeping this condition or not does not significantly impact the generation quality.

where $\hat{\pi}_{\text{inv}}(a|s) = \mathbb{E}_{s'}[\hat{f}(a|s, s')]$ denotes the empirical inverse policy, which is a marginal of inverse dynamics over $s'$.

Then the distance of generated data distribution to the optimal one is bounded as:

**Theorem 1** (Distribution shaping error bound). *Suppose that the optimal stationary state distribution satisfies that 1) its score function $\nabla_s \log d^*(s)$ is L-Lipschitz and 2) its second momentum is bounded. Under Assumption 1 and 2, the gap of generated state-action distribution to the optimal stationary state-action distribution is bounded by*

$$D_{\text{TV}}\left(d_g(s, a)\|d^*(s, a)\right) \leq \tilde{\mathcal{O}}\left(\varepsilon_{score}\sqrt{K}\right) + \sqrt{\varepsilon_{inv}/2} + C(d^*(s), L, K), \tag{12}$$

*where $C(d^*(s), L, K)$ represents a constant determined by $d^*(s)$, $L$ and $K$.*

The proof is given in Appendix A.2. Theorem 1 indicates that using the proposed OASIS method, we can shape the dataset distribution towards a bounded neighborhood of the optimal distribution.

Given the generated data, we will then train a regularized offline safe RL policy by Eq. (3). Notice that the regularization term in the objective function in Eq. (3) is equivalent to an explicit policy constraint, and the coefficient $w$ is the corresponding dual variable. Therefore, we make the following assumption on the distance between the learned policy $\pi_\phi$ and the behavior policy.

**Assumption 3.** Denote the generated dataset as $\mathcal{D}_g$ and the corresponding behavior policy as $\pi_g$, given a fixed coefficient $w$, for the policy $\pi_\phi$ optimized by Eq. (3), there exists a $\varepsilon_{\text{reg}}$ such that

$$\mathbb{E}_{d_g(s)}\left[D_{\text{KL}}\left(\pi_\phi(\cdot|s)\|\pi_g(\cdot|s)\right)\right] \leq \varepsilon_{\text{reg}}. \tag{13}$$

Based on the above assumptions, we can derive the bound of constraint violation of the policy learned on the offline data generated by OASIS. The proof is given in Appendix A.3.

**Theorem 2** (Constraint violation bound). *For policy $\pi_\phi$ optimized by regularized-based offline safe RL on generated dataset $\mathcal{D}_g$, under Assumption 1 , 2 and 3, the constraint violation of the trained policy is bounded as:*

$$V_c^{\pi_\phi}(\mu_0) - \kappa \leq \frac{2C_{max}}{1 - \gamma}\left(\tilde{\mathcal{O}}\left(\varepsilon_{score}\sqrt{K}\right) + C(d^*(s), L, K) + \sqrt{\varepsilon_{inv}/2} + \sqrt{\varepsilon_{reg}/2}\right), \tag{14}$$

*where $C(d^*(s), L, K)$ represents a constant determined by $d^*(s)$, $L$ and $K$.*

The theoretical analysis sheds insights by answering two questions: (1) Why do we use the diffusion model for conditional distribution shaping, and (2) How does conditional distribution shaping benefit offline safe RL training? Theorem 1 shows that by using the conditional diffusion model as a data generator, the TV distance between optimal and generated state-action distribution is bounded; Theorem 2 shows that the safety performance of agent trained on the generated dataset is guaranteed with OASIS.

## 5 Experiment

In the experiments, we answer these questions: (1) How does the distribution of the dataset influence the performance of regularized offline safe RL? (2) How does our proposed distribution shaping method perform in offline safe RL tasks? (3) How well does the conditional data generator shape the dataset distribution? To address these questions, we set up the following experiment tasks.

**Environments.** We adopt the continuous robot locomotion control tasks in the public benchmark `Bullet-Safety-Gym` [96] for evaluation, which is commonly used in previous works [68, 28, 60]. We consider two tasks, `Run` and `Circle`, and three types of robots, `Ball`, `Car`, and `Drone`. We name the environment as `Agent-Task`. A detailed description is available in the Appendix B.

**Datasets.** Our experiment tasks are mainly built upon the offline safe RL dataset `OSRL` [74]. To better evaluate the tested algorithms with the challenging SDM problem, we create four different training dataset types, `full`, `tempting`, `conservative`, and `hybrid`. The `tempting` dataset contains sparse safe demonstrations, the `conservative` dataset lacks rewarding data points, and the `hybrid` dataset has scarcity in the medium-reward, medium-cost trajectories. We set different cost thresholds for different datasets. A detailed description and visualization of the datasets are available in Appendix B. For the main experiments presented in Table 1, we train on `tempting` dataset, with threshold $\kappa = 20$.

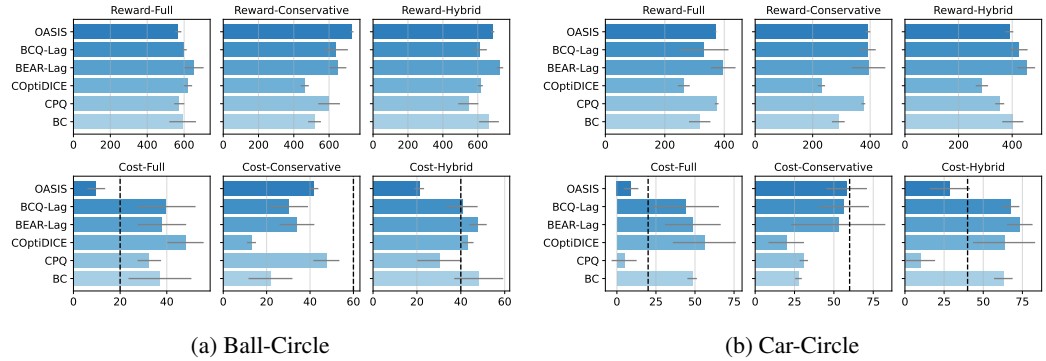

(a) Ball-Circle             (b) Car-Circle

Figure 5: Performance with different datasets and varying constraint thresholds. The visualization of these datasets is available in Appendix B.

**Baselines and OASIS.** For baselines, we compared our method with both *model-centric* and *data-centric* baseline methods. *Model-cenrtic*: (1) Q-learning-based algorithms: `BCQ-Lag` [35, 86], `BEAR-Lag` [41, 86], and `CPQ` [26]; (2) Imitation learning: Behavior Cloning (`BC`) [26]; (3) Distribution correction estimation: `COptiDICE` [23], and (4) Sequential modeling algorithms: `CDT` [28] and `FISOR` [31]; *Data-cenrtic*: (5) Data augmentation: `CVAE-BCQL`: we train BCQ-Lag agents on the datasets generated by Conditional Variational Autoencoder (CVAE) [97]. The `CVAE` training set and the dataset generation conditions are set as the same with our OASIS. For `OASIS` implementation, we set three cost conditions $\hat{C} \leq \kappa$ to align safety preference and guarantee data coverage. Code is available on our Github repository, checkpoints and curated datasets are available on our HuggingFace repository.

**Metrics.** We use the normalized cost return and the normalized reward return as the evaluation metric for comparison in Table 1 and 2. The normalized cost is defined as $C_{\text{normalized}} = C_\pi / \kappa$, where $C_\pi$ is the cost return and $\kappa$ is the cost threshold. The agent is safe if $C_{\text{normalized}} \leq 1$. The normalized reward is computed by $R_{\text{normalized}} = R_\pi / r_{\max}(\mathcal{M})$, where $r_{\max}(\mathcal{M})$ is the maximum empirical reward return for task $\mathcal{M}$ within the given dataset. We report the averaged results and standard deviations over 3 seeds for all the quantity evaluations.

Table 1: Evaluation results of the normalized reward and cost. The cost threshold is 1. ↑: the higher the reward, the better. ↓: the lower the cost (up to threshold 1), the better. **Bold**: Safe agents whose normalized cost is smaller than 1. Gray: Unsafe agents. **Blue**: Safe agent with the highest reward.

| Algorithm | Stats | Tasks | | | | | |
|---|---|---|---|---|---|---|---|
| | | BallRun | CarRun | DroneRun | BallCircle | CarCircle | DroneCircle |
| BC | reward ↑ | $0.55 \pm 0.23$ | $0.94 \pm 0.02$ | $0.62 \pm 0.11$ | $0.73 \pm 0.05$ | $0.59 \pm 0.11$ | $0.82 \pm 0.01$ |
| | cost ↓ | $2.04 \pm 1.32$ | $1.50 \pm 1.11$ | $3.48 \pm 0.68$ | $2.53 \pm 0.15$ | $3.39 \pm 0.85$ | $3.29 \pm 0.18$ |
| CPQ | reward ↑ | $0.25 \pm 0.11$ | $0.63 \pm 0.51$ | $0.13 \pm 0.30$ | $\mathbf{0.39 \pm 0.34}$ | $\mathbf{0.64 \pm 0.02}$ | $0.01 \pm 0.02$ |
| | cost ↓ | $1.34 \pm 1.32$ | $1.43 \pm 1.82$ | $2.29 \pm 1.98$ | $\mathbf{0.73 \pm 0.66}$ | $\mathbf{0.12 \pm 0.19}$ | $3.16 \pm 3.85$ |
| COptiDICE | reward ↑ | $0.63 \pm 0.04$ | $\mathbf{0.90 \pm 0.03}$ | $0.71 \pm 0.01$ | $0.73 \pm 0.02$ | $0.52 \pm 0.01$ | $\mathbf{0.35 \pm 0.02}$ |
| | cost ↓ | $3.13 \pm 0.17$ | $\mathbf{0.28 \pm 0.24}$ | $3.87 \pm 0.08$ | $2.83 \pm 0.23$ | $3.56 \pm 0.16$ | $\mathbf{0.12 \pm 0.10}$ |
| BEAR-Lag | reward ↑ | $0.65 \pm 0.08$ | $0.55 \pm 0.62$ | $0.10 \pm 0.33$ | $0.89 \pm 0.02$ | $0.80 \pm 0.08$ | $0.89 \pm 0.04$ |
| | cost ↓ | $4.38 \pm 0.28$ | $8.44 \pm 0.62$ | $3.72 \pm 3.22$ | $2.84 \pm 0.28$ | $2.89 \pm 0.84$ | $4.03 \pm 0.51$ |
| BCQ-Lag | reward ↑ | $0.51 \pm 0.19$ | $0.96 \pm 0.06$ | $0.76 \pm 0.07$ | $0.76 \pm 0.04$ | $0.79 \pm 0.02$ | $0.88 \pm 0.04$ |
| | cost ↓ | $1.96 \pm 0.88$ | $2.31 \pm 3.22$ | $5.19 \pm 1.08$ | $2.62 \pm 0.29$ | $3.25 \pm 0.28$ | $3.90 \pm 0.55$ |
| CDT | reward ↑ | $0.35 \pm 0.01$ | $\mathbf{\textcolor{blue}{0.96 \pm 0.01}}$ | $0.84 \pm 0.12$ | $0.73 \pm 0.01$ | $0.71 \pm 0.01$ | $0.17 \pm 0.08$ |
| | cost ↓ | $1.56 \pm 1.10$ | $\mathbf{\textcolor{blue}{0.67 \pm 0.03}}$ | $7.56 \pm 0.33$ | $1.36 \pm 0.03$ | $2.39 \pm 0.15$ | $1.08 \pm 0.62$ |
| FISOR | reward ↑ | $\mathbf{0.17 \pm 0.03}$ | $\mathbf{0.85 \pm 0.02}$ | $0.44 \pm 0.14$ | $\mathbf{0.28 \pm 0.03}$ | $\mathbf{0.24 \pm 0.05}$ | $\mathbf{0.49 \pm 0.05}$ |
| | cost ↓ | $\mathbf{0.04 \pm 0.06}$ | $\mathbf{0.15 \pm 0.20}$ | $2.52 \pm 0.61$ | $\mathbf{0.00 \pm 0.00}$ | $\mathbf{0.15 \pm 0.27}$ | $\mathbf{0.02 \pm 0.03}$ |
| CVAE-BCQL | reward ↑ | $0.25 \pm 0.02$ | $0.88 \pm 0.00$ | $0.21 \pm 52.07$ | $0.49 \pm 0.03$ | $0.60 \pm 0.05$ | $0.01 \pm 0.02$ |
| | cost ↓ | $1.40 \pm 0.35$ | $\mathbf{0.00 \pm 0.00}$ | $2.80 \pm 0.63$ | $1.39 \pm 0.27$ | $1.77 \pm 0.47$ | $3.31 \pm 1.66$ |
| OASIS (ours) | reward ↑ | $\mathbf{\textcolor{blue}{0.28 \pm 0.01}}$ | $0.85 \pm 0.04$ | $\mathbf{\textcolor{blue}{0.13 \pm 0.08}}$ | $\mathbf{\textcolor{blue}{0.70 \pm 0.01}}$ | $\mathbf{\textcolor{blue}{0.76 \pm 0.03}}$ | $\mathbf{\textcolor{blue}{0.60 \pm 0.01}}$ |
| | cost ↓ | $\mathbf{\textcolor{blue}{0.79 \pm 0.37}}$ | $0.02 \pm 0.03$ | $\mathbf{\textcolor{blue}{0.79 \pm 0.54}}$ | $\mathbf{\textcolor{blue}{0.45 \pm 0.14}}$ | $\mathbf{\textcolor{blue}{0.89 \pm 0.59}}$ | $\mathbf{\textcolor{blue}{0.25 \pm 0.10}}$ |

## 5.1 How can conditional data generation benefit offline safe RL?

**Performance degradation with SDM problems.** The comparison results on the `tempting` dataset are presented in Table 1 with the cost threshold $\kappa = 20$ before normalization. Results of `BC` show the mismatch between the behavior policy and the safe policy, as the cost returns significantly violate the safety constraints. The results of `BCQ-Lag` and `BEAR-Lag` show this mismatch further influences the regularized-based algorithms, leading to constraint violations. This is because the regularization term pushes the policy towards the unsafe behavior policy. The conservative Q function estimation method `CPQ`, exhibits a significant reward degradation in all tasks, which arises from the drawback of the pessimistic estimation methods that learn over-conservative behaviors. `COptiDICE` also fails to learn safe and rewarding policies, showing that even using distribution correction estimation is not enough to solve the SDM problem. For sequential modeling algorithms, `CDT` shows poor safety performance and `FISOR` tends to be over-conservative with poor reward performance. This is because both methods require a large amount of high-quality data while the trajectories with low cost and high reward are sparse in this task. The unsatisfactory performance of these *model-centric* algorithms further motivates the effective *data-centric* learning algorithm for offline safe RL.

**Performance improvement using OASIS.** From Table 1, we find that only our method OASIS can learn safe and rewarding policies by mitigating the SDM problem. In addition to the results on the `tempting` dataset, we also provide evaluation results within different types of datasets and constraint thresholds in Fig. 5a and Fig. 5b. We can observe that most baselines still fail to learn a safe policy within different task conditions due to the SDM issue. In contrast, our proposed OASIS method achieves the highest reward among all safe agents, which shows strength in more general cases.

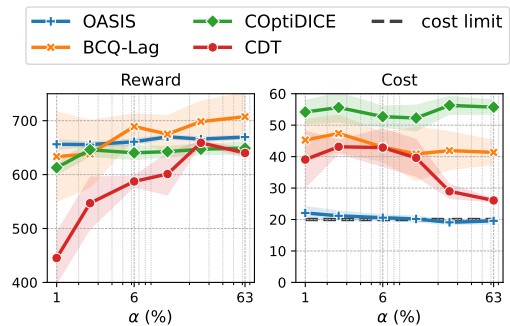

Figure 6: Data efficiency on the `Ball-Circle` task with a `tempting` dataset.

**High data efficiency of OASIS.** In this experiment, we vary the amount of data for offline RL agent training to evaluate data efficiency. The evaluation results are shown in Fig. 6. The x-axis $\alpha$ represents the size of the RL agent training dataset. For OASIS, it denotes the size of the generated data. A subsequent BCQ-Lag agent is trained on this generated dataset to obtain the safe RL policy. For the baseline methods, we randomly sample trajectories from the original dataset to construct the training dataset.

The comparison results indicate that we can still learn a good policy using a small amount of high-quality data ($\alpha < 2\%$) generated by OASIS. In contrast, baseline methods show significant performance degradation when the data are sparse as the noisy data is of low quality.

This observation demonstrates that the agent can learn a good policy with high data efficiency given high-quality data with minimal safe dataset mismatch (SDM) issues. As OASIS offers a solution to shape the dataset distribution, it also reduces the required training dataset size while maintaining good performance, further

Table 2: Ablation study on denoising step $K$

| | $K$ | 10 | 20 | 40 |
|---|---|---|---|---|
| Ball-Circle | reward | $0.71 \pm 0.02$ | $0.70 \pm 0.01$ | $0.71 \pm 0.01$ |
| | cost | $0.72 \pm 0.10$ | $0.45 \pm 0.14$ | $0.99 \pm 0.13$ |
| Ball-Run | reward | $0.29 \pm 0.04$ | $0.28 \pm 0.01$ | $0.29 \pm 0.01$ |
| | cost | $0.16 \pm 0.14$ | $0.79 \pm 0.37$ | $0.00 \pm 0.00$ |

reaffirming the effectiveness of *data-centric* approaches in offline safe learning and highlighting that prioritizing quality is essential [70].

## 5.2 How can OASIS shape the dataset distribution?

**Successful distribution shaping.** To show the distribution shaping capability of the proposed OASIS and baseline `CVAE`, we generate the dataset under different conditions and visualize them in Fig. 7. When using different conditions, the expectations of reward and cost of the generated dataset change accordingly. This shows the strong capability of our method in distribution shaping. We also visualize the density of the generated data. In the `Car-circle` task, the robot receives high rewards when moving along the circle boundary and receives costs when it exceeds the boundaries on both sides, as shown in Fig. 7(c). The original dataset contains trajectories with various safety performances.

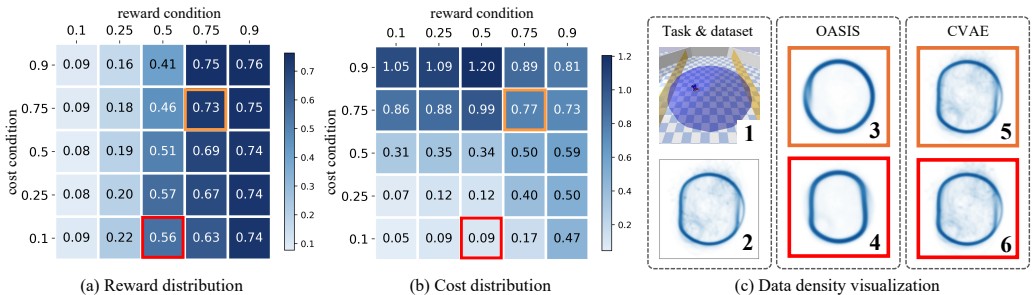

Figure 7: (a)(b) Reward and cost performance of the generated data: $\mathbb{E}\left[r(s,a)\right], \mathbb{E}\left[c(s,a)\right], (s,a) \sim d_g$. The x-axis and y-axis mean the reward and cost conditions and the values of both conditions and expectations are normalized with the same scale. (c) Visualization of the data density. 1: `Car-circle` task; 2: The density of the $(x,y)$ position of the raw dataset; 3, 4: The density of the position $(x,y)$ of the data generated by OASIS under conditions $[0.1, 0.5]$ and $[0.75, 0.75]$; 5, 6: the density of the position $(x,y)$ of the data generated by `CVAE` under conditions $[0.1, 0.5]$ and $[0.75, 0.75]$

When using a low-cost condition, the generated data are clustered within the safety boundary to satisfy the constraints. When using a high-reward condition, the generated data points are closer to the circle boundary and receive higher rewards. In contrast, the baseline `CVAE` cannot successfully incorporate the conditions in data generation, resulting in almost similar datasets with different conditions as shown in Fig. 7(c). More experiment results including the visualization of generated dataset comparison and corresponding analysis are available in Appendix B.2.

### 5.3 Robust performance against denoising steps

We conduct an ablation study on the key hyperparameter of the proposed OASIS method. The experiment related to the denoising steps $K$ is presented in Table 2. Performance does not change much with different values, which shows the robustness of the proposed OASIS method.

## 6 Conclusion

In this paper, we study the challenging problem in offline safe RL: the safe data mismatch between the imperfect demonstration and the target performance requirements. To address this issue, we proposed the OASIS method to employ a conditional diffusion model to shape the dataset distribution and benefit offline safe RL training. In addition to the theoretical guarantee of performance improvement, we also conduct extensive experiments to show the superior performance of OASIS in learning a safe and rewarding policy on many challenging offline safe RL tasks. More importantly, our method shows good data efficiency and robustness to hyperparameters, which makes it preferable for applications in many real-world tasks.

There are two limitations of OASIS: (1) Offline training takes longer: our method involves preprocessing the offline dataset to enhance quality, which requires more time and computing resources; (2) Achieving zero-constraint violations remains challenging with imperfect demonstrations. One potential negative social impact is that misuse of our method may cause harmful consequences and safety issues. Nevertheless, we believe that our proposed method can inspire further *data-centric* research in the safe learning community and help to adapt offline RL algorithms to real-world tasks with safety requirements.

## Acknowledgment

The work is partially supported by Google Deepmind with an unrestricted grant. The authors also want to acknowledge the support from the National Science Foundation under grants CNS-2047454.

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

# Appendix

## Table of Contents

## A  Proofs

### A.1  Proof of Eq.(9)

By definition of the stationary state-action distribution,

$$|V_c^\pi(\mu_0) - V_c^*(\mu_0)| \tag{15}$$

$$= \frac{1}{1-\gamma} \left| \mathbb{E}_{(s,a)\sim d^\pi}[c(s,a)] - \mathbb{E}_{(s,a)\sim d^*}[c(s,a)] \right| \tag{16}$$

$$\leq \frac{C_{\max}}{1-\gamma} \sum_{s,a} |d^\pi(s,a) - d^*(s,a)| \tag{17}$$

$$= \frac{2C_{\max}}{1-\gamma} D_{\mathrm{TV}}(d^\pi(s,a)\|d^*(s,a)) \tag{18}$$

$$\leq \frac{2C_{\max}}{1-\gamma} \left( D_{\mathrm{TV}}(d^\pi(s,a)\|d^*(s)\pi(a|s)) + D_{\mathrm{TV}}(d^*(s)\pi(a|s)\|d^*(s,a)) \right) \tag{19}$$

$$= \frac{2C_{\max}}{1-\gamma} D_{\mathrm{TV}}(d^\pi(s)\|d^*(s)) + \frac{2C_{\max}}{1-\gamma} \mathbb{E}_{d^*(s)}[D_{\mathrm{TV}}(\pi(a|s)\|\pi^*(a|s))] \tag{20}$$

The second inequality holds by triangle inequality for total variation distance.

In general, the stationary distribution of learned policy is in between of the empirical distribution of offline data $d^\mathcal{D}$ and optimal $d^*$. Therefore, we can obtain

$$|V_c^\pi(\mu_0) - V_c^*(\mu_0)| \leq \frac{2C_{\max}}{1-\gamma} D_{\mathrm{TV}}(d^*(s)\|d^\mathcal{D}(s)) + \frac{2C_{\max}}{1-\gamma} \mathbb{E}_{d^*(s)}[D_{\mathrm{TV}}(\pi(a|s)\|\pi^*(a|s))]. \tag{21}$$

### A.2  Proof of Theorem 1

*Proof.* By triangle inequality, we first decompose the TV distance between state-action distributions into a state distribution distance and a policy distance,

$$D_{\mathrm{TV}}\left(d_g(s,a)\|d^*(s,a)\right) \tag{22}$$

$$= D_{\mathrm{TV}}\left(d_g(s)\pi_g(a|s)\|d^*(s)\pi^*(a|s)\right) \tag{23}$$

$$\leq D_{\mathrm{TV}}\left(d_g(s)\pi_g(a|s)\|d^*(s)\pi_g(a|s)\right) + D_{\mathrm{TV}}\left(d^*(s)\pi_g(a|s)\|d^*(s)\pi^*(a|s)\right) \tag{24}$$

$$= D_{\mathrm{TV}}\left(d_g(s)\|d^*(s)\right) + \mathbb{E}_{d^*(s)}[D_{\mathrm{TV}}(\pi_g(a|s)\|\pi^*(a|s))] \tag{25}$$

We then consider two parts separately.

For the stationary state distribution distance, we suppose that the optimal distribution $d^*(s)$ has a $L$-Lipschitz smooth score function and a bounded second momentum. Meanwhile, note that the score function in Assumption 1 is closely related to the denoising model $\epsilon_\theta$ [98, 99]:

$$\nabla_s \log d_g(s|\boldsymbol{y}) = -\frac{1}{\sqrt{1-\bar{\alpha}_t}}\epsilon_\theta(s|\boldsymbol{y}), \tag{26}$$

where $\epsilon_\theta(s|\boldsymbol{y})$ is the state marginal of the practical denoising model in Eq. (8). Therefore, by theorem 2 in [95], under Assumption 1, we have

$$D_{\text{TV}}(d_g(s)\|d^*(s)) \lesssim \sqrt{D_{\text{KL}}(d^*(s)\|\mathcal{N}(\mathbf{0}, \mathbf{I}^{|\mathcal{S}|}))}\exp(-K) + L(\sqrt{|\mathcal{S}|}+\mathbf{m}_2)\sqrt{K} + \varepsilon_{\text{score}}\sqrt{K} \tag{27}$$

where $K$ is the number of denoising timestep, $|\mathcal{S}|$ is the dimension of the state space, and $\mathbf{m}_2$ is the second momentum of $d^*(s)$. Therefore, aggregating the first two terms in RHS, we have

$$D_{\text{TV}}(d_g(s)\|d^*(s)) \leq \tilde{\mathcal{O}}\left(\varepsilon_{\text{score}}\sqrt{K}\right) + C(d^*(s), L, K), \tag{28}$$

where $C(\dots)$ is a constant w.r.t $d^*, L, K$.

Regarding the policy distance. By Pinsker's inequality,

$$D_{\text{TV}}(\pi_g(a|s)\|\pi^*(a|s)) \leq \sqrt{\frac{1}{2}D_{\text{KL}}(\pi_g(a|s)\|\pi^*(a|s))} \tag{29}$$

Meanwhile, since the action is generated by the inverse policy, i.e., $\pi_g = \hat{\pi}_{\text{inv}}$, by Assumption 2, we have

$$\mathbb{E}_{d^*(s)}\left[D_{\text{TV}}(\pi_g(a|s)\|\pi^*(a|s))\right] \leq \mathbb{E}_{d^*(s)}\left[\sqrt{\frac{1}{2}D_{\text{KL}}(\pi_g(a|s)\|\pi^*(a|s))}\right] \tag{30}$$

$$\leq \sqrt{\frac{1}{2}\mathbb{E}_{d^*(s)}\left[D_{\text{KL}}(\pi_g(a|s)\|\pi^*(a|s))\right]} \tag{31}$$

$$= \sqrt{\varepsilon_{\text{inv}}/2} \tag{32}$$

where the second inequality holds by Jensen's inequality.

Combining the Eq. (28)) and (32), we finish the proof of Theorem 1. $\square$

### A.3 Proof of Theorem 2

We start from the Eq. (9). The policy distance can be further decomposed into

$$D_{\text{TV}}(\pi(a|s)\|\pi^*(a|s)) \leq D_{\text{TV}}(\pi(a|s)\|\pi^{\mathcal{D}}(a|s)) + D_{\text{TV}}(\pi^{\mathcal{D}}(a|s)\|\pi^*(a|s)). \tag{33}$$

By Assumption 2 and 3 and Jensen's inequality, we have

$$\mathbb{E}_{d^*(s)}[D_{\text{TV}}(\pi(a|s)\|\pi^*(a|s))] \tag{34}$$

$$\leq \mathbb{E}_{d^*(s)}[D_{\text{TV}}(\pi(a|s)\|\pi^{\mathcal{D}}(a|s))] + \mathbb{E}_{d^*(s)}[D_{\text{TV}}(\pi^{\mathcal{D}}(a|s)\|\pi^*(a|s))] \tag{35}$$

$$\leq \mathbb{E}_{d^*(s)}\left[\sqrt{D_{\text{KL}}(\pi(a|s)\|\pi^{\mathcal{D}}(a|s))/2}\right] + \mathbb{E}_{d^*(s)}\left[\sqrt{D_{\text{KL}}(\pi^{\mathcal{D}}(a|s)\|\pi^*(a|s))/2}\right] \tag{36}$$

$$\leq \sqrt{\mathbb{E}_{d^*(s)}\left[D_{\text{KL}}(\pi(a|s)\|\pi^{\mathcal{D}}(a|s))\right]/2} + \sqrt{\mathbb{E}_{d^*(s)}\left[D_{\text{KL}}(\pi^{\mathcal{D}}(a|s)\|\pi^*(a|s))\right]/2} \tag{37}$$

$$\leq \sqrt{\varepsilon_{\text{reg}}/2} + \sqrt{\varepsilon_{\text{inv}}/2} \tag{38}$$

Plug-in Eq. (28) and (38) into Eq. (21), we have

$$|V_c^\pi(\mu_0) - V_c^*(\mu_0)| \leq \frac{2C_{\text{max}}}{1-\gamma}\left(\tilde{\mathcal{O}}\left(\varepsilon_{\text{score}}\sqrt{K}\right) + C(d^*(s), L, K) + \sqrt{\varepsilon_{\text{inv}}/2} + \sqrt{\varepsilon_{\text{reg}}/2}\right). \tag{39}$$

Meanwhile, notice that the optimal policy is constraint satisfactory, i.e.,

$$V_c^*(\mu_0) \leq \kappa. \tag{40}$$

Therefore, we have

$$V_c^\pi(\mu_0) - \kappa \leq \frac{2C_{\text{max}}}{1-\gamma}\left(\tilde{\mathcal{O}}\left(\varepsilon_{\text{score}}\sqrt{K}\right) + C(d^*(s), L, K) + \sqrt{\varepsilon_{\text{inv}}/2} + \sqrt{\varepsilon_{\text{reg}}/2}\right), \tag{41}$$

which finishes the proof of Theorem 2.

# B  Supplementary experiments

## B.1  Trajectory reweighting for distribution shaping

In this section, we provide details about the trajectory reweighting experiment presented in Sec. 4.1. Following previous work [39, 89] in offline RL, we adopted datapoint reweighting in policy optimization, which can be formulated via importance sampling as:

$$\mathcal{J}_{\text{off}}^{w}(\pi, \lambda) \approx \mathbb{E}_{(s,a)\sim\mathcal{D}_w}[\mathcal{J}_{\text{off}}(\pi, \lambda)] = \mathbb{E}_{(s,a)\sim\mathcal{D}}[w(s,a)\mathcal{J}_{\text{off}}(\pi, \lambda)], \tag{42}$$

where $\mathcal{J}_{\text{off}}^{w}(\pi, \lambda)$ is the objective function after reweighting. In this experiment, we utilize a Boltzmann energy function as adopted in [39] for offline RL tasks:

$$w(\tau) \propto \exp\left(\alpha_1 R(\tau)^2 + \alpha_2 (C(\tau) - \kappa)^2\right) \tag{43}$$

Here $w(\tau)$ means that all the state-action pairs in one trajectory share the same weight, which is related to the cost and reward returns $C(\tau), R(\tau)$. We adopt $\alpha_1 = \alpha_2 = 1$ in the experiments shown in Fig. 3.

## B.2  Supplementary data generation comparison results

Due to the page limit, we omit the visualization of reward and cost distribution using the CVAE method for data generation. Here we provide the results in Fig. 8. From the reward performance and the cost performance, we can observe that CVAE can hardly encode conditions into the data reconstruction, leading to similar results when setting different conditions. From the trajectory reconstruction results shown in Fig. 8(c), we can observe that the generated trajectories are almost the same as the original one. This feature is not desirable for our distribution shaping purpose. In contrast, our method OASIS can successfully shape the distribution as shown in Fig. 7, with the strong capability of the diffusion model in the condition-guided denoising process.

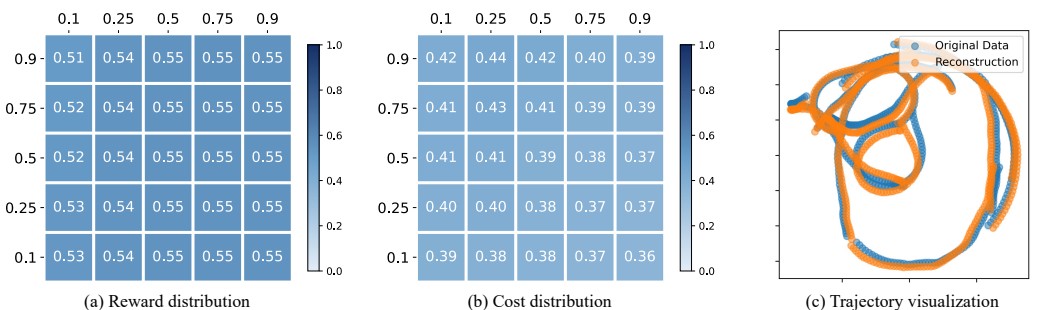

Figure 8: CVAE reconstruction. (a) Reward performance of the generated data: $\mathbb{E}\left[r(s,a)\right], (s,a) \sim d_g$, (b) Cost performance of the generated data: $\mathbb{E}\left[c(s,a)\right], (s,a) \sim d_g$. In (a) and (b), the x-axis and y-axis mean the reward and cost conditions, and the value of both conditions and expectations are normalized to the same scale: $[0, 1]$; (c) The data reconstruction results using the condition $[0.1, 0.5]$ of 10 sampled trajectories in the dataset.

The comparison results of generated trajectories using OASIS and `CVAE` in the `Drone-Circle` task are presented in Fig. 9. This figure illustrates the generation results of OASIS and CVAE under two conditions: low-cost-medium-reward, and medium-cost-high-reward. Although CVAE can reconstruct the trajectories, it fails to integrate conditions into the generation process. In contrast, OASIS successfully controls the generated trajectories, avoiding the restricted area when conditioned on low-cost-medium-reward.

# C  Implementation details

## C.1  Environment details

Due to the page limit, we omit some descriptions of experiments in the main context. Here we give more details about our experiment tasks. Both the Circle task and the Run task are from a publicly available benchmark [96].

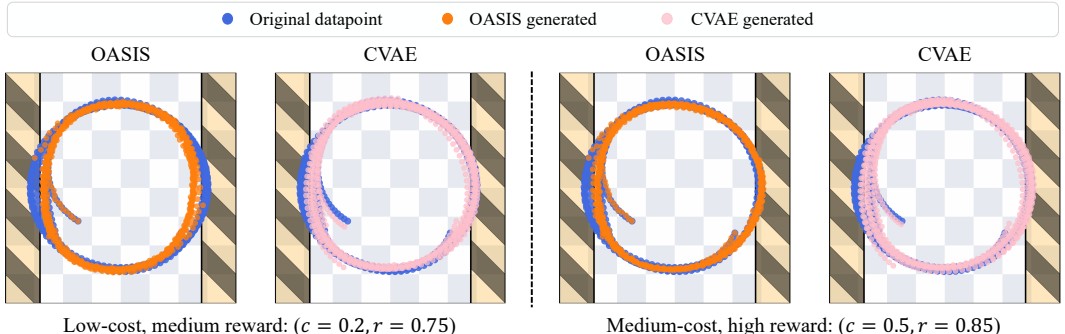

Figure 9: Visualization of generated trajectories in the `Drone-Circle` task.

**Circle tasks.** The agents are rewarded for running along a circle boundary. The reward function is defined as:

$$r(s, a, s') = \frac{-yv_x + xv_y}{1 + |\sqrt{x^2 + y^2} - radius|} + r_{robot}(s) \tag{44}$$

where $x$, $y$ are the positions of the agent with state $s'$, $v_x$, and $v_y$ are velocities of the agent with state $s'$. $radius$ is the radius of the circle area, and $r_{robot}(\boldsymbol{s_t})$ is the specific reward for different robot.

The agent gets cost when exceeding the boundaries. The cost function is defined as:

$$\text{Boundary: } c(\boldsymbol{s_t}) = \mathbf{1}(|x| > x_{\lim}) \tag{45}$$

where $x_{lim}$ is the boundary position.

**Run tasks.** Agents are rewarded for running fast along one fixed direction and are given costs if they run across the boundaries or exceed a velocity limit. The reward function is defined as:

$$r(s, a, s') = ||\boldsymbol{x_{t-1}} - \boldsymbol{g}||_2 - ||\boldsymbol{x_t} - \boldsymbol{g}||_2 + r_{robot}(s_t) \tag{46}$$

The cost function is defined as:

$$c(s, a, s') = \max(1, \mathbf{1}(|y| > y_{lim}) + \mathbf{1}(||\boldsymbol{v_t}||_2 > v_{lim})) \tag{47}$$

where $v_{lim}$ is the speed limit, and $y_{lim}$ is the $y$ position of the boundary, $\boldsymbol{v_t} = [v_x, v_y]$ is the velocity of the agent with state $s'$, $\boldsymbol{g} = [g_x, g_y]$ is the position of a virtual target, $\boldsymbol{x_t} = [x_t, y_t]$ is the position of the agent at timestamp $t$, $\boldsymbol{x_{t-1}}$ is the Cartesian coordinates of the agent with state $s$, $\boldsymbol{x_t}$ is the Cartesian coordinates of the agent with state $s'$, and $r_{robot}(\boldsymbol{s_t})$ is the specific reward for the robot.

**Agents.** We use three different robot agents in our experiments: `Ball`, `Car`, and `Drone`. The action space dimension, observation space dimension, and the timesteps for these six tasks are shown in Table 3.

Table 3: Environment description

|  | Max timestep | Action space dimension | Observation space dimension |
|---|---|---|---|
| BallRun | 100 | 2 | 7 |
| CarRun | 200 | 2 | 7 |
| DroneRun | 200 | 4 | 17 |
| BallCircle | 200 | 2 | 8 |
| CarCircle | 300 | 2 | 8 |
| DroneCircle | 300 | 4 | 18 |

## C.2 Dataset details

We provide details about the dataset types we presented in the experiment part. The `Full`, `Tempting`, `Conservative`, and `Hybrid` datasets for `Ball-Circle` and `Car-Circle` tasks are shown in Fig. 10, 11, respectively. All the `Tempting` datasets associated with results in Table 1 are shown in Fig. 12.

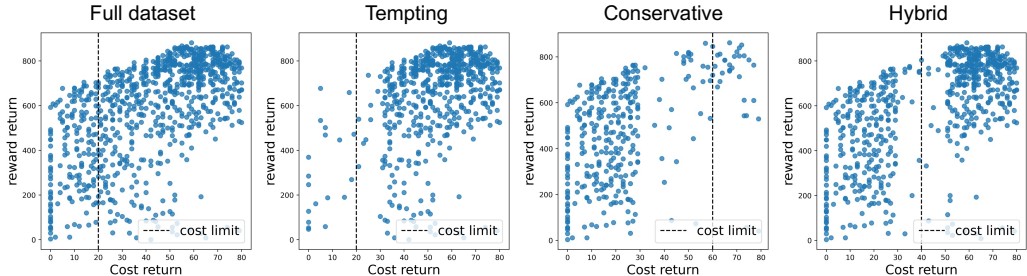

Figure 10: BallCircle Dataset types. Each point represents $(C(\tau), R(\tau))$ of a trajectory $\tau$ in the dataset.

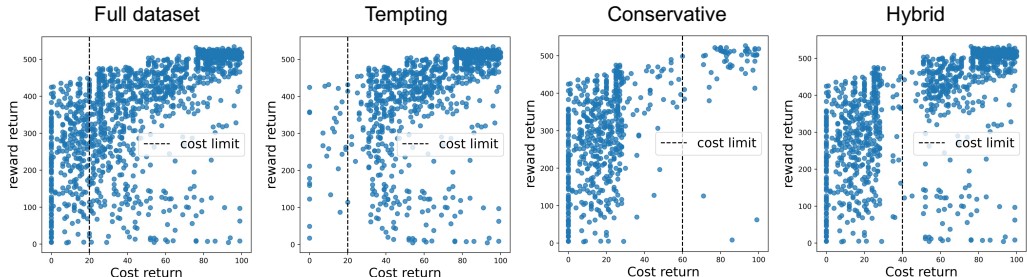

Figure 11: CarCircle Dataset types. Each point represents $(C(\tau), R(\tau))$ of a trajectory $\tau$ in the dataset.

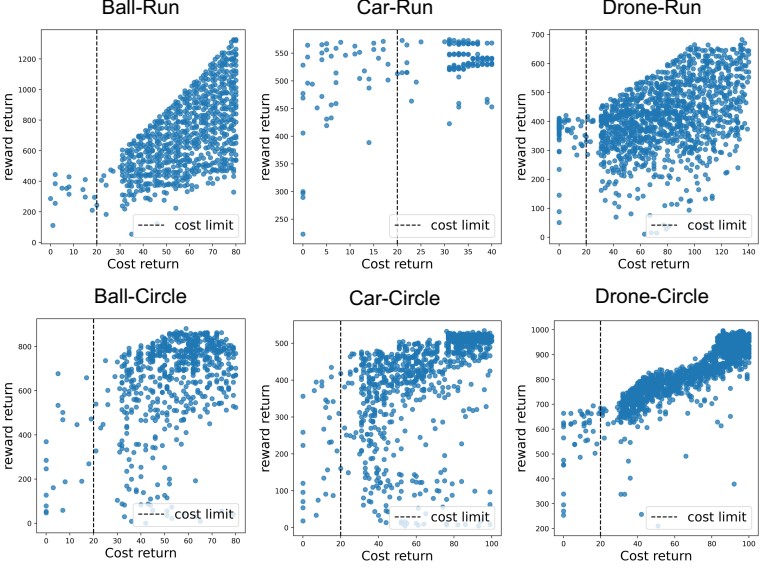

Figure 12: All tempting datasets. Each point represents $(C(\tau), R(\tau))$ of a trajectory $\tau$ in the dataset.

## C.3  Algorithm details

**OASIS algorithm training diagram**  In this work, we use the cosine $\beta$ schedule [100] to calculate $\beta_t, t = 1, ..., K$. Then we let $\alpha_t = 1 - \beta_t$, and $\bar{\alpha}_t = \prod_{i=1}^{t} \alpha_i$ and denote the state dimension as $m$. With these notations, we show the training process of the OASIS data generator for one epoch in Algorithm 2.

---

**Algorithm 2:** OASIS (training)

---

1: **Input:** Original Dataset $\mathcal{D}$, predefined $\beta_t$ and $\bar{\alpha}_t$, diffusion core $\epsilon_\theta\,(\boldsymbol{x}_k, \boldsymbol{y}, k)$, learning rate $lr$, loss function $L(\cdot, \cdot)$.
2: **for** each sub-trajectory $\tau_i \in \mathcal{D}$ **do**
3:     Extract the states from $\tau_i$: $\{(s_0, s_1, \ldots, s_T)\}$ # $[T, m]$;
4:     Get the return conditions $\boldsymbol{y} = [C, R]$ associated with these sub-trajectories;
5:     With probability $p = 0.25$ to mask the condition information as: $\boldsymbol{y} \leftarrow \varnothing$
6:     Get Gaussian Noise $noise = \mathcal{N}(\boldsymbol{0}, \boldsymbol{I})$ # $[T, m]$;
7:     Randomly sample time $t \in [0, ..., K-1]$;
8:     Calculate the forward sampling state $\boldsymbol{x}_{noise} = \bar{\alpha}_t * \tau_i + (1 - \bar{\alpha}_t) * noise$;
9:     Apply initial state condition $\boldsymbol{x}_{noise}[0] \leftarrow s_0$;
10:    Reconstruct noisy sub trajectory $\boldsymbol{x}_{recon} = \epsilon_\theta\,(\boldsymbol{x}_{noise}, \boldsymbol{y}, k)$;
11:    Minimize the reconstruction loss $\theta \leftarrow \theta - lr * \nabla_\theta L(\boldsymbol{x}_{noise}, \boldsymbol{x}_{recon})$;
12: **end for**
13: **Output:** Updated diffusion core $\epsilon_\theta\,(\boldsymbol{x}_k, \boldsymbol{y}, k)$;

---

**Network and hyperparameter details.**  For the dynamics model $\hat{p}$, we utilize a MLP. For the denoising core, we utilize a `U-net`, which has also been used in previous works [82, 80]. The `U-net` is visualized in Fig. 13. The hyperparameters for our method are summarized in Table 4. More details are available in the code.

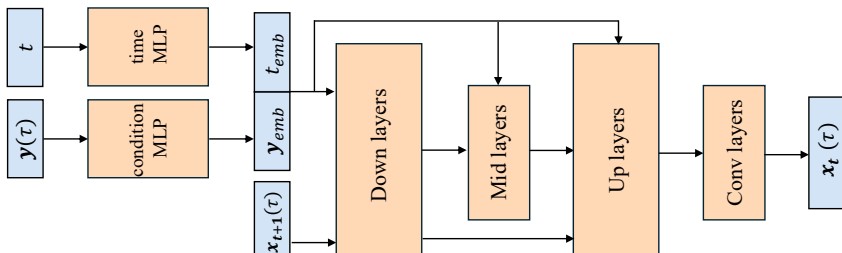

Figure 13: `U-Net` structure.

**Baseline details.**  For the baseline methods `BC`, `BCQ-Lag`, `BEAR-Lag`, `COptiDICE`, `CPQ`, and `CDT`, we adopt the code base provided in the benchmark [74]. For the `FISOR` method, we use the code provided by the authors [31].

**Computing resources.**  The experiments are run on a server with $2\times$AMD EPYC 7542 32-Core Processor CPU, $2\times$NVIDIA RTX A6000 graphics, and 252 GB memory. For one single experiment, OASIS takes about 4 hours with $200,000$ steps to train the data generator. It takes about $1.5$ hours to train a `BCQ-Lag` agent on this generated dataset for $200,000$ steps.

Table 4: Hyperparameters

| Hyperparameters | Value |
|---|---|
| L (length of subsequence) | 32 |
| K (denoising timestep) | 20 |
| Batch size | 256 |
| Learning rate | 3.0e-5 |
| $w_\alpha$ | 2.0 |

