# OpenReview forum: "OASIS: Conditional Distribution Shaping for Offline Safe Reinforcement Learning"
_NeurIPS.cc/2024/Conference — NeurIPS 2024 poster_

### Official Review · Reviewer_XmVn · 2024-06-19

**Soundness:** 3
**Presentation:** 3
**Contribution:** 3
**Rating:** 6
**Confidence:** 4

**Summary:**

This paper discusses the safe dataset mismatch (SDM) problem, highlighting how low-reward or unsafe samples in datasets can harm offline safe RL. Conditional distribution shaping (OASIS) is proposed to mitigate this problem by generating high-reward and safe samples via diffusion models and promoting general offline safe RL algorithms with the generated data. This paper evaluates OASIS through extensive experiments across various safe RL tasks and different types of datasets with varying data distributions.

**Strengths:**

- The paper is well motivated. The influence of imbalance and biased data on offline safe RL is an important but underexplored problem. Solving the proposed problem through diffusion-based data generation is intuitive, reasonable and novel.
- The empirical evaluation and ablation studies are comprehensive, carefully demonstrating the significance of the SDM problem and the effectiveness of the proposed approach.
- Theoretical analysis provides certain guarantees for the proposed approach.
- The paper is well written and well organized.

**Weaknesses:**

- Some technical and experimental details are a bit confusing. See Questions 1, 2, 3, 4.

- Assumptions 1 and 2 seem kind of idealized to directly bound the distribution and policy discrepancy, since $\epsilon_{score}$ and $\epsilon_{inv}$ cannot be directly calculated or estimated. Admittedly, analyzing the distribution and policy discrepancy based on diffusion-generated data may be difficult and beyond the scope of this work. Maybe more discussion and explanation could help justify these assumptions.

- Directly excluding mismatched data from training datasets could be an another intuitive approach for SDM problem. So it would be better to discuss the performance if tempting and conservative data were directly excluded from the full datasets, utilizing the remaining datasets (i.e., Full dataset - Hybrid dataset) for training. A comparison between these two approaches (i.e., adding matched data vs. reducing mismatched data) would be valuable.

**Questions:**

1. What is the size of the generated dataset $D_g$? I would like to confirm whether you directly train the policy on $D_g$ or a mixture of $D_g$ and the original datasets?
2. How do you calculate the reward condition $\hat R$ based on the given cost $\hat C$?
3. Why are the thresholds in Figure 5 different for different types of datasets? Did you use the same threshold when constructing different types of datasets? This is a concern because one tempting dataset constructed under the thresholds 20 may be considered tempting for one threshold 20 but not for another threshold 60.
4. In Figure 6, how did you obtain the performance of baseline methods (e.g., CDT) under a specific $\alpha$? Did you use the data of size $\alpha$ generated by OASIS for these baselines? It would be helpful to provide more details about the data efficiency experiments.

**Limitations:**

The limitation are discussed in the paper.

---

> ### Author Rebuttal · Authors · 2024-08-06
>
> We gratefully thank the reviewer for recognizing the novelty, comprehensive experiment validation, and theoretical analysis contribution of our work. We provide our response to the comments below:
>
> > W2: Assumptions 1 and 2 seem kind of idealized to directly bound the distribution and policy discrepancy, since and cannot be directly calculated or estimated. Admittedly, analyzing the distribution and policy discrepancy based on diffusion-generated data may be difficult and beyond the scope of this work. Maybe more discussion and explanation could help justify these assumptions.
>
> Assumption 1 and 2 bound the function approximation error of diffusion model and inverse dynamics model, which depends on the implementation (e.g., network architecture, learning rate, etc.). As you mentioned, we also believe it may be out of the scope to directly analyze this type of error. Meanwhile, those assumptions bound the expectation of error instead of maximal error and they are also adopted by previous work [1,2]. Therefore, we argue that the assumptions are not overly idealized but necessary to derive theoretical analysis.
>
> > W3: Directly excluding mismatched data from training datasets could be an another intuitive approach for SDM problem. So it would be better to discuss the performance if tempting and conservative data were directly excluded from the full datasets, utilizing the remaining datasets (i.e., Full dataset - Hybrid dataset) for training. A comparison between these two approaches (i.e., adding matched data vs. reducing mismatched data) would be valuable.
>
> According to the reviewer's suggestion, we conduct the experiments with these datasets in the Ball-Circle task with threshold 40. The used hybrid, hybrid-R(rewarding only), hybrid-S(safe-only), and hybrid-Com(complementary) datasets are visualized in Figure R-4 presented in the supplementary PDF file. We can observe that when using the safe-only dataset, the learned policy is conservative with low cost and low reward. When using the rewarding-only dataset, the learned policy is tempting with high reward and high cost. When training with the complement of the hybrid dataset, we can get a better safe performance compared to using the hybrid dataset. However, since it still contains a large amount of imperfect demonstration with low reward, the reward performance is not satisfactory compared to our OASIS.
>
> |Stats|OASIS|BCQ-Lag (hybrid)|BCQ-Lag (hybrid-R)|BCQ-Lag (hybrid-S)|BCQ-Lag (hybrid-Com)|
> |-|-|-|-|-|-|
> |Reward|$684.87\pm9.66$|$672.99\pm40.54$|$780.91\pm15.41$|$594.99\pm27.75$|$646.98\pm70.60$|
> |Cost ($\kappa=40$)|$32.10\pm3.27$|$52.11\pm1.84$|$63.39\pm2.00$|$24.17\pm5.06$|$33.77\pm9.23$|
>
> > Q1: What is the size of the generated dataset? I would like to confirm whether you directly train the policy on or a mixture of and the original datasets?
>
> The number of transition pairs of the original tempting dataset and the generated datasets used in Table 1 is shown below. Each value presents the number of transition pairs.
>
> ||BallCircle|CarCircle|DroneCircle|BallRun|CarRun|DroneRun|
> |-|-|-|-|-|-|-|
> |full dataset|177200|435000|576243|94000|130200|395099|
> |tempting dataset|126200|302100|426705|73900|29600|249432|
> |OASIS generated $D_g$|62000|62000|155000|62000|62000|155000|
>
> For OASIS, we directly train RL agents on the generated datasets.
>
> > Q2: How do you calculate the reward condition based on the given cost?
>
> We select the conditions with similar study in Figure 7 to find a good condition that has a high reward while sticking to the cost limit.
>
> > Q3: Why are the thresholds in Figure 5 different for different types of datasets? Did you use the same threshold when constructing different types of datasets? This is a concern because one tempting dataset constructed under the thresholds 20 may be considered tempting for one threshold 20 but not for another threshold 60.
>
> We change the thresholds in Figure 5 to show that our method can perform well under varying threshold conditions. We used different thresholds to construct different types of datasets to make them tempting/conservative/hybrid. The visualization of the dataset and corresponding threshold is also available in Appendix C.2.
>
> According to the reviewer's suggestion, we also conducted experiments with the same thresholds on different datasets. The datasets and the experiment results are presented in Figures R-2 and R-3 in the provided PDF file. We can observe that when setting threshold=40, in all tested datasets including tempting, conservative, and hybrid, our method OASIS exhibits the best performance, achieving the highest reward among the safe agents.
>
> > Q4: In Figure 6, how did you obtain the performance of baseline methods (e.g., CDT) under a specific $\alpha$? Did you use the data of size generated by OASIS for these baselines? It would be helpful to provide more details about the data efficiency experiments.
>
> We apologize for any confusion. In Figure 6, $\alpha$ represents the size of the RL agent training dataset. For OASIS, it denotes the size of the generated data. A subsequent BCQ-Lag agent is trained on this generated dataset to obtain the safe RL policy. For the baseline methods, we create the training dataset by randomly sampling $\alpha\%$ of trajectories from the original dataset for RL training. In this experiment, we aim to demonstrate that, for offline safe RL, the agent can learn a good policy in a data-efficient manner if the dataset has minimal safe dataset mismatch (SDM) issues. OASIS offers a solution to shape the dataset distribution, which can reduce the required dataset size for RL training while maintaining good performance. We have added a detailed explanation and analysis of this experiment in the revision.
>
> ---
>
> [1] Holden Lee, et al. Convergence for score-based generative modeling with polynomial complexity.
>
> [2] Sitan Chen, et al. Sampling is as easy as learning the score: theory for diffusion models with minimal data assumptions.

---

> > ### Comment · Reviewer_XmVn · 2024-08-12
> >
> > Thanks for the authors' response. The supplementary experiments and clarifications are comprehensive and help address my concerns. I maintain my score in favor of accepting the paper.

---

### Official Review · Reviewer_fA23 · 2024-07-09

**Soundness:** 3
**Presentation:** 4
**Contribution:** 3
**Rating:** 7
**Confidence:** 4

**Summary:**

Offline Safe Reinforcement Learning (RL) is used to learn policies satisfying cost constraints from a given dataset. This proves to be a challenge when the dataset is biased in a certain way. This paper introduces a method to use the offline training dataset to capture the environment using a diffusion model that can generate data conditioned on our cost constraints and performance objectives. Experimental results show the approach generates data that better fit the constraints and outperform alternatives like dataset reweighting.

**Strengths:**

- The paper is well organized and clearly shows the strengths of diffusion models to generate offline data with the given cost/performance objectives.
- Theoretical results show that the policies learned from the generated data are cost constrained given some reasonable assumptions on the model.
- Extensive comparisons are made to different SoTA baselines in offline RL with and without data generation.

**Weaknesses:**

- Data generation and training can be a slow process due to the use of diffusion models which are known to be computationally heavy.
- Selecting hyperparameters such as number of denoising steps might vary results significantly. For example, Table 2 has greatly varying policy costs for different values of $K$ (albeit similar performances).
- While results are mostly consistent, it is hard to say when the proposed method prefers to act more conservatively with lower cost (or more riskily i.e., higher reward). This is reflected in the reward and costs in Table 1 (e.g., CarRun). A study on a hyperparameter change (apart from cost/return targets) to control this balance would be helpful.

**Questions:**

1. How are the initial states decided to generate the data from the diffusion model? Are they the same as the initial states of the training dataset or randomly sampled from the training dataset trajectories?
2. How long is inference time for the OASIS model i.e., data generation time? Is that included in the training time (L999, App C.3)?
3. How do we decide the “target” cost given the cost threshold to get the best performance? Is the study done in Fig. 7 required for each setting or is there a good heuristic to set these targets? How expensive is this study (Fig. 7) over target values (i.e., multiple inference runs)? On a related note, how were these targets set for Table 1 on OASIS and the compared baselines?
4. Is it right to say the primary reason for the success of the proposed method over the diffusion baselines (like FISOR), a more realistic conditional generative model that yields optimal generated data satisfying our cost constraints?
5. Are the learned labeling models (inverse dynamics, reward, and cost) only used for labeling the data generated from the diffusion model?
6. In Fig. 6, are all models using data generated by the same diffusion model? OASIS handles the diffusion model training and data generation with BCQ-Lag as the actual Offline RL policy learning. This makes me a little confused. How are the curves related in Fig. 6?
7. Why is hybrid an interesting dataset setting vs. full?
8. Typos:
    - L224 (Sec 4.4) Theoretical
    -  L973 (App C.1) min instead of max

**Limitations:**

Diffusion models being computationally heavy (see Weakness)

---

> ### Author Rebuttal · Authors · 2024-08-06
>
> We appreciate the reviewer for the insightful suggestions and the praise of our theoretical results and experimental performance. We provide our response to the reviewer's comments and questions as follows:
>
> > W1: Data generation and training can be a slow process.
>
> We agree that generating new data could cause extra computational cost to the training process. However, since we focus on the offline RL setting, this extra cost only occurs before online inference, which we believe is still acceptable.
>
> > W2: Selecting hyperparameters such as the number of denoising steps might vary results significantly. For example, Table 2 has greatly varying policy costs for different values of $K$.
>
> For offline safe RL tasks, agents sometimes achieve similar rewards while obtaining different cost values when the cost limit is small and the cost signal is binary. The difference in cost performance in Table 2, where the threshold is set at $\kappa=20$, may come from this. Nevertheless, our method consistently outperforms the baselines in terms of safety constraint satisfaction and reward maximization.
>
> We also tested our method with moderate thresholds ($\kappa=30, 40$) and show the results below. We observe that the cost performance of our method is more consistent under this setting. (All the values are normalized.)
>
> **BallCircle ($\kappa=30$):**
>
> |$K$|10|20|40|
> |-|-|-|-|
> |Reward|$0.793\pm0.014$|$0.777\pm0.017$|$0.791\pm0.010$|
> |Cost|$0.877\pm0.182$|$0.857\pm0.123$|$0.970\pm0.099$|
>
> **BallCircle ($\kappa=40$):**
>
> |$K$|10|20|40|
> |-|-|-|-|
> |Reward|$0.809\pm0.007$|$0.783\pm0.015$|$0.808\pm0.014$|
> |Cost|$0.845\pm0.121$|$0.740\pm0.059$|$0.819\pm0.041$|
>
> **BallRun ($\kappa=30$):**
>
> |$K$|10|20|40|
> |-|-|-|-|
> |Reward|$0.317\pm0.025$|$0.341\pm0.031$|$0.318\pm0.032$|
> |Cost|$0.890\pm0.514$|$0.739\pm0.385$|$0.797\pm0.361$|
>
> **BallRun ($\kappa=40$):**
>
> |$K$|10|20|40|
> |-|-|-|-|
> |Reward|$0.374\pm0.019$|$0.392\pm0.047$|$0.364\pm0.011$|
> |Cost|$0.896\pm0.137$|$0.825\pm0.336$|$0.963\pm0.176$|
>
> > W3: While results are mostly consistent, it is hard to say when the proposed method prefers to act more conservatively with lower cost. A study on a hyperparameter change (apart from cost/return targets) to control this balance would be helpful.
>
> In our method, the balance between reward and cost of generated data for the same model is only controlled by the input condition, i.e., the target reward/cost.
>
> To further study the influence of other hyperparameters on the learned generative model, we provide more ablation experiments on the sequence length $L$. We use the Ball-Circle task with a threshold of $\epsilon=30$ and show results on the testing dataset in the following table. We find that our method is not sensitive to $L$ within the tested range.
>
> |$L$|32|48|64|
> |-|-|-|-|
> |Reward|$0.799\pm0.014$|$0.803\pm0.005$|$0.785\pm0.014$|
> |Cost|$0.890\pm0.132$|$0.926\pm0.090$|$0.798\pm0.070$|
>
> > Q1: How are the initial states decided to generate the data from the diffusion model?
>
> They are randomly sampled from the training dataset trajectories.
>
> > Q2: How long is the inference time for the OASIS model? Is that included in the training time?
>
> The generation time in all experiments is less than 1 minute with an A6000 GPU. Taking Ball-Circle as an example, we train the diffusion model of OASIS with a sequence length of $L=32$. During generation, we randomly sample 2000 states from the training dataset. After one-time generation, we obtain a dataset with $2000 \times (32-1) = 62000$ transitions. This process only takes 11 seconds. The generation time is not included in the training time presented in L999, App C.3.
>
> > Q3: How do we decide the “target” cost given the cost threshold to get the best performance? Is the study done in Fig. 7 required for each setting or is there a good heuristic to set these targets? How expensive is this study (Fig. 7)? On a related note, how were these targets set for Table 1 on OASIS and the compared baselines?
>
> We first set the “target” cost as the normalized cost threshold, then adjust the conditions according to the results in Figure 7 to find a good condition that has a high reward while adhering to the cost limit. It is a one-time inference for one set of conditions, which only takes about 15 seconds in total. Meanwhile, the performance of each condition is evaluated by the learned reward and cost and does not require online data. For CDT, which requires an additional reward condition, the reward conditions are adopted from the source code released by the authors.
>
> > Q4: Is it right to say the primary reason for the success of the proposed method over the diffusion baselines (like FISOR), is a more realistic conditional generative model that yields optimal generated data satisfying our cost constraints?
>
> In the offline RL setting, the distribution of the dataset matters a lot. If the data quality is low, i.e., the dataset is imbalanced and mostly contains unsafe or low-rewarding demonstrations, our OASIS method, which shapes the dataset distribution towards the target distribution, is more effective than baselines (e.g., FISOR) directly modeling the policy by conditional generative model.
>
> > Q5: Are the learned labeling models (inverse dynamics, reward, and cost) only used for labeling the data generated from the diffusion model?
>
> Yes, they are only used to label the data generated from the diffusion model.
>
> (continued)

---

> ### Author Response · Authors · 2024-08-06
> **Continued rebuttal**
>
> (continued)
>
> > Q6: In Fig. 6, are all models using data generated by the same diffusion model? OASIS handles the diffusion model training and data generation with BCQ-Lag as the actual Offline RL policy learning. This makes me a little confused. How are the curves related in Fig. 6?
>
> We apologize for any confusion. In Figure 6, $\alpha$ represents the size of the RL agent training dataset. For OASIS, it denotes the size of the generated data. A subsequent BCQ-Lag agent is trained on this generated dataset to obtain the safe RL policy. For the baseline methods, we create the training dataset by randomly sampling $\alpha\%$ of trajectories from the original dataset for RL training. In this experiment, we aim to demonstrate that, for offline safe RL, the agent can learn a good policy in a data-efficient manner if the dataset has minimal safe dataset mismatch (SDM) issues. OASIS offers a solution to shape the dataset distribution, which can reduce the required dataset size for RL training while maintaining good performance. We have added a detailed explanation and analysis of this experiment in the revision.
>
> > Q7: Why is a hybrid dataset an interesting setting vs. a full dataset?
>
> The hybrid dataset is more realistic in some real-world tasks. For example, in autonomous driving, we define the cost as the distance to the nearest surrounding obstacle and the reward as the time to arrive at the destination. In data collection, some conservative drivers achieve low cost but medium rewards. Some aggressive drivers achieve high reward but high cost. The combination of these two results in a hybrid dataset.
>
> > Q8: typos.
>
> We fixed these typos in our revised version and carefully checked the manuscript.

---

> > ### Comment · Reviewer_fA23 · 2024-08-12
> >
> > I thank the authors for their response and have no further questions.

---

### Official Review · Reviewer_CUpS · 2024-07-11

**Soundness:** 2
**Presentation:** 2
**Contribution:** 2
**Rating:** 6
**Confidence:** 2

**Summary:**

This paper proposes OASIS, which uses a conditional diffusion model to reshape the dataset distribution and achieve effective offline safe RL learning. Theoretical analysis gives the error upper bound of distribution reshaping and constraint violation upper bound. A large number of experiments show that the proposed algorithm has significantly improved compared to offline safe RL baselines.

**Strengths:**

- The logic is clear and the paper is easy to understand.
- The theoretical analysis is sufficient.
- A large number of experiments prove the effectiveness of the proposed algorithm.

**Weaknesses:**

***W1:*** Some baselines are not introduced in related work, such as CVAE, FISOR, etc. This may cause difficulties in understanding.

***W2:*** Typos, such as line 173 "reweighing" -> "reweighting"; wrong citations, such as line 290 "COptiDICE [17]" -> "COptiDICE [16]"

**Questions:**

***Q1:*** Compared with CVAE, OASIS shows that the conditional diffusion model has a better ability to generate according to the condition information, as shown in Figure 7(c), but this example is a bit simple. Can the author show more comparisons of the two similar to Figure 7(c)? For example, add the OASIS generation results to Figure 8(c)?

***Q2:*** When showing the effectiveness of the newly generated dataset, in addition to showing the distribution of the generated state like Figure 7(c), it is also necessary to verify whether the annotations of the inverse dynamics model and reward & cost models are accurate. Can the author supplement the accuracy of these three models?

***Q3:*** By comparing the results of OASIS and CDT, I came to a conclusion: in offline safe RL, both methods conditioned on cost and reward, generating data is more effective than generating policy. I wonder if the author agrees with this conclusion? This conclusion seems to be uncommon in the field of offline RL, or it may be because I am not familiar with distribution shaping methods.

**Limitations:**

See the weaknesses and questions sections.

---

> ### Author Rebuttal · Authors · 2024-08-06
>
> We would like to express our gratitude to the reviewer for the valuable feedback. We are glad to know that the reviewer recognizes the clear logic, sufficient theoritical analysis, and experiments proving the effectiveness. We provide our response to the questions and concerns below.
>
> > W1: Some baselines are not introduced in related work, such as CVAE, FISOR, etc. This may cause difficulties in understanding.
>
> We thank the reviewer for pointing this out. We have added the following sentences to the related work section. FISOR identifies the largest feasible region where agents can operate safely, and optimizes for high rewards within this region and minimizes risks outside it with a diffusion policy. CVAE is an extension of the standard VAE that incorporates additional conditional information, such as class labels or attributes, into data generation. This enables more controlled and specific output generation.
>
> > W2: Typos.
>
> Thanks for your careful review. we fixed these typos in our revised version.
>
> > Q1: Compared with CVAE, OASIS shows that the conditional diffusion model has a better ability to generate according to the condition information, as shown in Figure 7 c, but this example is a bit simple. Can the author show more comparisons of the two similar to Figure 7 c? For example, add the OASIS generation results to Figure 8 c?
>
> We present additional comparisons in Figure R-1 of the supplementary PDF file. This figure illustrates the generation results of OASIS and CVAE under two conditions: medium-cost-high-reward and low-cost-medium-reward. While CVAE can reconstruct the trajectories, it fails to integrate conditions into the generation process. In contrast, OASIS successfully controls the generated trajectories, avoiding the restricted area when conditioned on low-cost-medium-reward.
>
> > Q2: When showing the effectiveness of the newly generated dataset, in addition to showing the distribution of the generated state like Figure c, it is also necessary to verify whether the annotations of the inverse dynamics model and reward & cost models are accurate. Can the author supplement the accuracy of these three models?
>
> We provided the scaled MSE loss values of inverse dynamics, reward and cost models in the following tables. We provide the inital value of the loss when the training begins for a comparison. The error of these models is small.
>
> **Inverse dynamics**
> ||BallCircle| CarCircle | DroneCircle | BallRun | CarRun | DroneRun |
> |- | - | - | - | -| - | - |
> |init | 1.08 | 1.04 | 1.58 | 0.68 | 1.14 | 1.03 |
> |final | 0.037 | 0.18 | 0.041 | 0.056 | 0.26 | 0.015 |
>
> **reward model**
> || BallCircle| CarCircle | DroneCircle | BallRun | CarRun | DroneRun |
> |- | - | - | - | - | - | - |
> |init | 1.86 | 7.89 | 1.54 | 3.96 | 1.96 | 1.92 |
> |final | 0.009 | 0.05 | 0.004  | 0.013 | 0.002 | 0.017 |
>
> **cost model**
> || BallCircle| CarCircle | DroneCircle | BallRun | CarRun | DroneRun |
> |- | - | - | - | - | - | - |
> |init | 5.59| 5.29  | 5.41 | 4.09 | 4.42 | 5.80 |
> |final | 0.13 | 0.15| 0.15 | 0.32 | 0.38 | 0.27 |
>
>
> > Q3: By comparing the results of OASIS and CDT, I came to a conclusion: in offline safe RL, both methods conditioned on cost and reward, generating data is more effective than generating policy. I wonder if the author agrees with this conclusion? This conclusion seems to be uncommon in the field of offline RL.
>
> In the offline RL setting, the distribution of datasets matters a lot. If the data quality is low, i.e. the dataset is imbalanced, and most contain unsafe or low-rewarding demonstrations, our OASIS, which shapes the dataset distribution towards the target distribution, is more effective than directly generating safe policy by conditional sequential modeling such as CDT. The importance of distribution shaping for imbalanced dataset in offline RL has also been discussed in some related works [1, 2], where the authors proposed sampling strategies to solve this issue.
>
> We acknowledge that the performance comparison between general policy generation and data generation in offline safe RL remains an open question. This represents a fascinating area for future research.
>
> ---
> [1] Hong, Zhang-Wei, et al. "Beyond uniform sampling: Offline reinforcement learning with imbalanced datasets." NeurIPS 2023.
>
> [2] Hong, Zhang-Wei, et al. "Harnessing mixed offline reinforcement learning datasets via trajectory weighting." ICLR 2024.

---

> > ### Comment · Reviewer_CUpS · 2024-08-13
> > **Response to Authors**
> >
> > Thank you for your response, which has addressed most of my concerns. However, I still have a few additional questions:
> >
> > ---
> >
> > Supplement to ***Q1***: As I mentioned earlier, the example of Car-Circle seems somewhat simplistic. If time permits, could you please add the generation results of OASIS to Figure 8c?
> >
> > Supplement to ***Q3***: Yes, I agree that the conclusion you provided is more accurate and reasonable.
> >
> > Based on the current discussions, I will at least maintain my score.

---

> > > ### Author Response · Authors · 2024-08-14
> > >
> > > We thank the reviewer for the useful and constructive comments. We answer the additional questions as follows.
> > >
> > > > Supplement to Q1: As I mentioned earlier, the example of Car-Circle seems somewhat simplistic. If time permits, could you please add the generation results of OASIS to Figure 8c?
> > >
> > > - Figure 8c shows the CVAE reconstruction results for the Car-Circle task, supplementing Figure 7c. During the rebuttal phase, we added the OASIS generation results to Figure 8c as requested by the suggestions:
> > >
> > >     > "_Q1: Compared with CVAE, OASIS shows that the conditional diffusion model has a better ability to generate according to the condition information, as shown in Figure 7c, but this example is a bit simple. Can the author show more comparisons of the two similar to Figure 7c? For example, add the OASIS generation results to Figure 8c?_"
> > >
> > > - The Figure R-1 we provided in the rebuttal phase is the similar generation results of OASIS to those of CVAE shown in Figure 8c. To make a clear visualization, we (1) reduce the number of trajectories; (2) add different conditions for generation; and (3) visualize the generation results from OASIS and CVAE in separate figures. The random seeds to sample trajectories are different, so the targets for reconstruction are slightly different in Figure R-1 and Figure 8c. However, we keep the same set of trajectories for reconstruction for OASIS and CVAE to make a clear and fair comparison in Figure R-1.
> > >
> > > - We select the Car-Circle task for this visualization experiment because (1) it is widely used in offline safe RL benchmark [1] and related offline safe RL works [2, 3, 4]; (2) its state space contains the position information of ego agent, which is easy to visualize in 2D space.
> > >
> > > - For the request to visualize other tasks, we will update the visualization results for other robots (i.e., Drone) that have high-dimensional observation and action space and complicated dynamics models. Since we can not update the PDF file at this stage, we will include these in the appendix of the revised manuscript.
> > >
> > > > Supplement to Q3: Yes, I agree that the conclusion you provided is more accurate and reasonable.
> > >
> > > We thank the reviewer for the agreement and acknowledgment. We have added related discussions in our revised manuscript.
> > >
> > >
> > > ---
> > > [1] Zuxin Liu, et al. "Datasets and benchmarks for offline safe reinforcement learning." arXiv preprint arXiv:2306.09303 (2023).
> > >
> > > [2] Yinan Zheng, et al. "Safe Offline Reinforcement Learning with Feasibility-Guided Diffusion Model." ICLR 2024
> > >
> > > [3] Zijian Guo, et al. "Temporal Logic Specification-Conditioned Decision Transformer for Offline Safe Reinforcement Learning." ICML 2024.
> > >
> > > [4] Kihyuk Hong, et al. "A primal-dual-critic algorithm for offline constrained reinforcement learning." AISTATS 2024.

---

> > > > ### Comment · Reviewer_CUpS · 2024-08-14
> > > > **Response to Authors**
> > > >
> > > > Thank you for your response. In the next version, please include the visual results of OASIS and CVAE in the Drone environment in the appendix. I greatly appreciate the additional experiments you conducted. Since you have addressed my concerns, I will raise the score to 6.

---

> > > > > ### Author Response · Authors · 2024-08-14
> > > > >
> > > > > Thanks again for all the insightful suggestions and comments. We will include the visual results of OASIS and CVAE for the Drone task in the appendix.

---

### Author Rebuttal · Authors · 2024-08-06

Dear Reviewers,

We thank you all for your careful review and valuable feedback. In addition to addressing each reviewer’s comments, we would like to highlight the new examples and experiments during the rebuttal phase. The figures we refer to can be found in the attached PDF file.

1. **Additional Visualization of Comparison Between OASIS and CVAE**
In response to reviewer CUpS, we present additional comparison results of trajectories generated by OASIS and CVAE in Figure R-1.

2. **Accuracy of the Learned Inverse Dynamics, Reward, and Cost Models**
In response to reviewer CUpS, we present the accuracy of these models.

3. **Additional Ablation Experiments**
In response to reviewer fA23, we present more results and analysis of our ablation study.

4. **Additional Experiments for Distribution Shaping Methods**
In response to reviewer XmVn, we conduct experiments to show the performance of the distribution shaping method by adding matched data and reducing mismatched data.

5. **Additional Experiments with Different Types of Datasets**
In response to reviewer XmVn, we construct different types of datasets (tempting/conservative/hybrid) under the same threshold and provide supplementary experiments.

We sincerely appreciate your time, attention, and valuable feedback.

Best regards,

Authors of Submission 12456

---

### Decision · Program_Chairs · 2024-09-25

**Decision:**

Accept (poster)

**Comment:**

One of the reviewers for this paper did not participate in the review process. The paper received relatively high scores: Reviewer CUpS (Score 6, Confidence 2), Reviewer fA23 (Score 7, Confidence 4), and Reviewer XmVn (Score 6, Confidence 4). As the AC, I took my responsibility seriously and carefully reviewed the paper. After the rebuttal, most of the reviewers' concerns were addressed. I thoroughly read through the communication between the authors and the reviewers, noting that the authors need to include some additional experiments in the revised version.

I also carefully reviewed the content of the paper. The main focus of the paper is the application of safe RL in the offline setting. However, I find the experimental setup to be overly simplistic, as it is limited to only two tasks: Run and Circle. Additionally, the introduction of the diffusion model leads to inefficiencies and hyperparameter issues, which the authors did not adequately resolve. Reviewers also pointed out that the theoretical assumptions are overly idealized.

Considering all the reviewers' opinions and my own thorough review as AC, this paper is accepted as a poster.